# Small molecule induced STING degradation facilitated by the HECT ligase HERC4

Merve Mutlu ®[1] ✉, Isabel Schmidt ®[1], Andrew I. Morrison ®[1,3],
Benedikt Goretzki[1], Felix Freuler[1], Damien Begue[1], Oliver Simic[1], Nicolas Pythoud[1],
Erik Ahrne[1], Sandra Kapps[1], Susan Roest[1], Debora Bonenfant[1,4],
Delphine Jeanpierre[1], Thi-Thanh-Thao Tran[1], Rob Maher[2], Shaojian An ®[2],
Amandine Rietsch[1], Florian Nigsch ®[1], Andreas Hofmann ®[1],
John Reece-Hoyes[2,5], Christian N. Parker[1] & Danilo Guerini[1]

Stimulator of interferon genes (STING) is a central component of the cytosolic nucleic acids sensing pathway and as such master regulator of the type I interferon response. Due to its critical role in physiology and its' involvement in a variety of diseases, STING has been a focus for drug discovery. Targeted protein degradation (TPD) has emerged as a promising pharmacology for targeting previously considered undruggable proteins by hijacking the cellular ubiquitin proteasome system (UPS) with small molecules. Here, we identify AK59 as a STING degrader leveraging HERC4, a HECT-domain E3 ligase. Additionally, our data reveals that AK59 is effective on the common pathological STING mutations, suggesting a potential clinical application of this mechanism. Thus, these findings introduce HERC4 to the fields of TPD and of compound-induced degradation of STING, suggesting potential therapeutic applications.

The cyclic GMP-AMP synthase (cGAS)-stimulator of interferon genes (STING, also known as TMEM173 or STING1) pathway is an important element of the innate immune system for the detection of aberrant intracellular DNA molecules. The pathway is activated by the binding of cGAS to cytosolic double-stranded deoxyribonucleic acid (DNA), thereby initiating the production of 2′,3′ cyclic GMP-AMP (cGAMP), a potent STING agonist. Located in the endoplasmic reticulum (ER) membrane in its naive state, STING dimerizes upon binding of cGAMP, translocates from the ER to the Golgi apparatus, and eventually triggers downstream interferon signaling through a cascade initiated by TANK binding kinase 1 (TBK1) phosphorylation followed by interferon regulatory factor 3 (IRF3) phosphorylation triggering the release of cytokines such as C-X-C motif chemokine ligand 10 (CXCL10) and interferon beta (IFNβ)[1]. The presence of double-stranded DNA in the cytoplasm is a hallmark of viral/bacterial infection, retroviral

replication, as well as cellular stress (which could also be caused by the release of mitochondrial DNA) and the production of cGAMP followed by STING activation, mediates as an innate immune response to such stimuli[2].

Involvement of the cGAS/STING pathway in various autoimmune diseases and cancer, has attracted significant interest for drug discovery in recent years[3]. While the activation of the pathway has been investigated as an important therapeutic tool for cancer immunotherapy[4,5], aberrant activation of the pathway has been associated with numerous autoimmune diseases[3], including ataxia-telangiectasia (AT), a genetic disorder linked to ATM kinase loss of function, as well as Aicardi- Goutières syndrome (AGS), a relatively rare inflammatory disease[6,7]. Moreover, STING hyperactivity, as the key regulator of the pathway, is associated with severe autoimmune diseases. Single-point mutations in STING are drivers of STING-associated

[1]Novartis BioMedical Research, Basel, Switzerland. [2]Novartis BioMedical Research, Cambridge, MA, USA. [3]Present address: Amsterdam UMC location Vrije Universiteit Amsterdam, Molecular Cell Biology & Immunology, Amsterdam institute for Infection and Immunity, De Boelelaan, 1117 Amsterdam, The Netherlands. [4]Present address: Monte Rosa Therapeutics, Basel, Switzerland. [5]Present address: Vector Biology, Cambridge, MA, USA. ✉e-mail: merve.koch@novartis.com

vasculopathy with onset in infancy (SAVI disease), a condition characterized by skin lesions, rashes, and interstitial lung disease[8] and thus antagonizing STING, e.g., through downregulation of STING expression, may have a high potential therapeutic value in the field of immunology[9]. While a straightforward approach to STING inhibition is the targeting of the cGAMP binding pocket[10] in the protein's active site, alternative strategies for STING inhibition are emerging, such as targeting post-translational modifications (PTMs)[11].

Ubiquitination is an important PTM that regulates numerous proteins in eukaryotic cells through the ubiquitin-proteasome system (UPS)[12]. It is achieved by a cascade of reactions involving ubiquitin-activating enzymes (E1), ubiquitin-conjugating enzymes (E2), and ubiquitin ligases (E3). In the first step, ubiquitin is activated via the E1 enzyme and, in the second step transferred to an E2 enzyme. The mechanism of the third step, in which ubiquitin is transferred to the substrate, is determined by the type of E3 ligase[13]. Really interesting new gene (RING)-finger containing the family of E3 ligases such as anaphase-promoting complex (APC), SKP1-Cullin1-F-box (SCF) complexes or DDB1-CUL4-X-box (DCX) complexes function as scaffolds mediating close proximity between the charged E2 and the target protein, thereby enabling ubiquitin transfer from E2 directly to the substrate[13–15]. By contrast, Homologous to the E6-AP Carboxyl Terminus (HECT)-domain-containing (HECT-type) E3 ligases receive ubiquitin from the E2 and transfer to the substrate themselves, utilizing an intermediate step of ubiquitin bound to their active site cysteine[16]. HERC4 is a member of the HECT-type E3 ligase family and is among the least well-characterized E3 ligases. Few reports have associated HERC4 expression with breast cancer, hepatocellular carcinoma, and lung tumors[17–23]. Additionally, only a few target substrates of HERC4 have been identified, such as Smoothened protein (SMO)[24,25], salvador (SAV) protein[26], c-Maf[27], and progesterone receptor (PGR)[28]. The structure and the mechanism of action of HERC4 have not yet been fully elucidated.

Interest in E3 ligases has increased tremendously with the discovery of molecular glue degraders (MGDs) and proteolysis-targeting chimera (PROTACs). These molecules can mediate the association of an E3 ligase to a target protein that is usually not a natural substrate of that E3 ligase, a so-called neosubstrate, thereby inducing targeted protein degradation (TPD) via the UPS. Such chemically induced protein–protein interactions (PPI) brought a potential paradigm of pharmacology to the field of drug discovery. While most TPD approaches rely on E3 ligases of the RING family, the potential of other ligase families for TPD, including HECT ligases, is yet to be explored.

In this work, we describe the characterization of the small molecule AK59-51TB (in short AK59) that inhibits the cGAS/STING pathway via degradation of STING. A genome-wide CRISPR-Cas9 screen identifies genes inhibiting this compound-mediated STING degradation. Subsequently, the function of AK59 is dependent on HERC4, UBA5, and UBA6, each of which reduces AK59 activity when individually knocked out. Moreover, immunoprecipitation assays and cell-based complementation PPI assay reveal the interaction between STING and HERC4 which only occurs in the presence of the compound. Thus, we present evidence that AK59 acts as a small molecule STING degrader, which subsequently leads to inhibition of the cGAS/STING pathway. The fact that AK59 acts through HERC4 demonstrates the potential of the HECT ligase family as potential ligases for TPD approaches.

## Results

### AK59 is an inhibitor of STING

Due to the therapeutical importance of the cGAS/STING pathway, we aimed to identify chemical modulators of STING activity, as one of the key regulators of the pathway. Initially, we screened a portion of the Novartis compound collection ($1.2 \times 10^6$ compounds) with a cell-based assay to identify potential agonists of STING activation. During the screening campaign, we discovered potential STING antagonists as well. To characterize the hits, Dual-THP1-Cas9 cells (that have a Lucia luciferase reporter linked to IRF responsive promoter) were used for hit characterization and also as an assay to define the structure-activity relationship (SAR) assays. Compound effects were monitored upon stimulation of the pathway with 30 μM cGAMP. Dose-response over cGAS/STING pathway activity, which is detected by luminescence, enabled us to identify AK59-51TB (in short AK59, Fig. 1a, b, see Supplementary Methods) which exhibited a significant inhibitory effect on cGAS/STING pathway activity. Strikingly, this inhibitory effect was substantially reduced for the close analog QK50-66NB (in short QK50, Fig. 1a, b, see Supplementary Methods).

Before we proceed with further functional characterization. we tested compound incubation time and dose to optimize for maximum effect with minimal toxicity. We conducted luminescence-based cell viability assays on THP1 cells treated with 10 μM, AK59 during the 0–72-h incubation periods (Supplementary Fig. 1a). A 16-h treatment window was selected as the longest incubation period without significant compound-mediated cell toxicity. At this point, we aimed to keep consistent dose and incubation times of compound treatment (10 μM for 16 h) for better comparability of results across experiments.

To further assess the inhibitory effect of AK59 on cGAS/STING pathway, we checked downstream phosphorylation of TBK1 and IRF3 by western blot as well as CXCL10 and IFNβ cytokine release upon STING stimulation by homogeneous time resolved fluorescence (HTRF) assay. While the DMSO control and QK50 treated Dual-THP1 cells showed IRF3 and TBK1 phosphorylation, AK59-treated cells did not, which is consistent with the observed degradation of STING, reducing its cellular abundance (Supplementary Fig. 1b). In parallel to these findings, CXCL10 and IFNβ release from cGAMP stimulated Dual-THP1 cells significantly decreased as AK59 concentration increases whereas in QK50 treated samples, cytokine levels stay relatively similar (Supplementary Fig. 1c, d). Together these findings are consistent with AK59 inhibiting STING-mediated activation of the IRF pathway.

In order to understand how AK59 antagonizes STING, we investigated STING expression levels upon AK59 treatment. For this purpose, we used THP1 cells and did not stimulate the cGAS/STING pathway via cGAMP. Strikingly, we observed STING expression level decrease upon increasing doses of AK59 while upon QK50 treatment remained stable (Fig. 1c). These finding let us to believe that STING is not only inhibited but also might be degraded by AK59. To further validate as well as use clinically relevant models, besides THP1 cells as our monocytic cell line model, we have also tested the function of AK59 (and QK50) on CD14 positive monocytes from human peripheral blood cells (PBMCs). Freshly isolated CD14+ PBMCs showed a significant decrease in STING expression upon 10 μM AK59 treatment (Fig. 1d and Supplementary Fig. 1e). Parallel to our THP1 findings, 10 μM QK50 treatment did not change STING expression. This showed us the AK59 function on STING degradation is translatable to primary tissue cultures.

One of the major protein degradation/recycling machineries in eukaryotic cells is UPS which controls protein levels of numerous cellular pathways[29]. To investigate the mechanism of action of AK59 in monocytic cells, we chose to use an unbiased proteomics approach. THP1 cells were treated with either DMSO (vehicle), 10 μM AK59, or 10 μM QK50 for 16 h alone or with an additional 50 nm bortezomib treatment. The treatment conditions were compared to the respective vehicle control group, and fold changes were calculated accordingly. The proteomics analysis revealed significantly downregulated proteins with a $\log_2$(fold change) $<-1$ and a false discovery rate $p$ value ($q$ value) of less than 0.01 (Fig. 1e and Supplementary Data 1). These proteomics approaches let us to identify potential targets of AK59 either proteasome-dependent or independent manner (Supplementary Data 1). Among these hits, STING expression levels were found to be significantly downregulated upon AK59 treatment compared to the DMSO control (Fig. 1e). Downregulation of STING in the proteomic

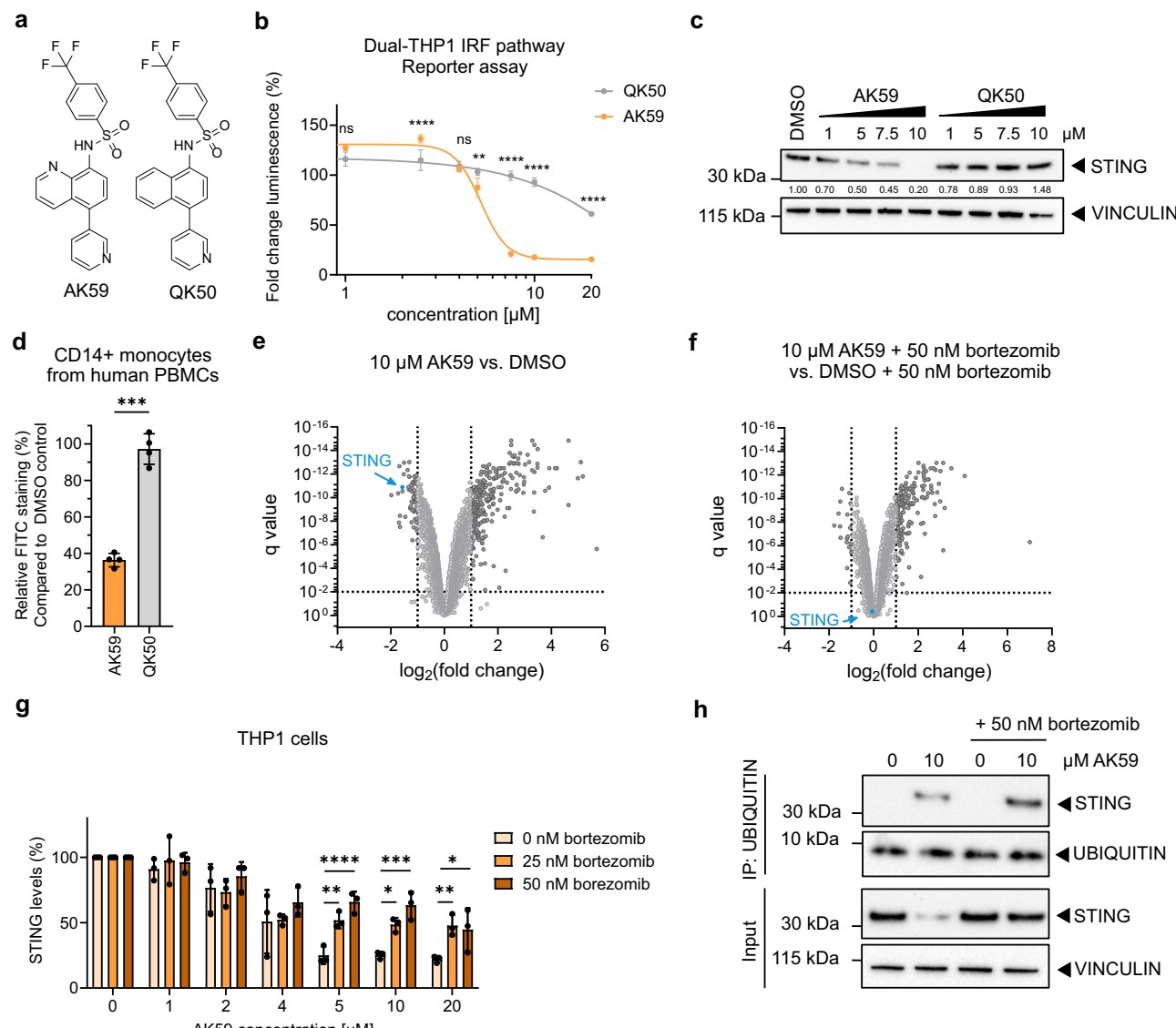

**Fig. 1 | AK59 induces STING depletion in THP1 cells through proteasomal degradation. a** Chemical structure of AK59 and QK50. **b** IRF pathway reporter assay on wild-type Dual-THP1 cells treated with increasing doses of AK59 or QK50 for 16 h. Data plotted as mean ± SD of three individual biological replicates. Calculated half-maximal inhibitory concentrations (IC50s) are 5.196 and 54.23 for AK59 and QK50 respectively. Significance was calculated using two-way ANOVA followed by Šidák's correction. Significance indicated as ns $p > 0.05$, **$p = 0.004$, ****$p < 0.0001$. **c** Western blot showing STING protein expressions in THP1 cells upon increasing doses of AK59 or QK50. Results are representative of three independent experiments. **d** STING expression of CD14+ human PBMCs treated with either 10 μM AK59 or QK50 over 16 h. Four donors were represented as biological replicates and plotted as mean ± SD as well as individual samples represented. Significance calculated using paired $t$-test. Significance indicated as ***$p = 0.0001$. **e**, **f** Proteomic analysis of 10 μM AK59-treated THP1 cells in the absence (**e**) or

presence (**f**) of bortezomib and compared to control DMSO control. Significantly altered protein abundances (in dark grey) are shown with a log2 fold change <− 1 or >1 and a $q$ value cutoff of 0.01. **g** STING expression of THP1 cells treated with AK59 with or without prior bortezomib treatment (25 or 50 nM). STING expression was detected via FACS analysis on fixed cells. Bortezomib treatment was 1 h prior to AK59 treatments. Each treatment group was normalized to its individual DMSO control and represented in percentage. Three biological replicates were plotted as the mean ± SD. Significance was calculated using two-way ANOVA followed by Šidák's correction. Significance indicated as ns $p > 0.05$, ** at 5 μM $p = 0.0068$, *** at 10 μM $p = 0.0001$, * at 10 μM $p = 0.0191$, ** at 20 μM $p = 0.0096$, * at 20 μM $p = 0.0237$, ***$p < 0.001$. **h** Ubiquitin pulldown followed by western blot on THP1 cells treated with 16 h of 10 μM AK59 and 1 h of our prior 50 nM bortezomib addition. Source data are provided as a Source Data file.

profile of AK59-treated THP1 cells was rescued upon co-treatment with 50 nM bortezomib (Fig. 1f), thus indicating a UPS-dependent STING degradation mechanism through AK59. STING expression levels remained unaltered in the presence of inactive analog QK50 compared to the DMSO control as well as the bortezomib-treated counterpart of the proteomics data (Supplementary Fig. 1g, h).

To further validate this finding, we analyzed STING protein levels in AK59-treated THP1 cells by western blot. Proteasomal degradation was inhibited using two different proteasome inhibitors (bortezomib

and MG132). In addition, a neddylation inhibitor (MLN4924) was tested on AK59-treated THP1 cells (Supplementary Fig. 1i) to test whether AK59-induced STING degradation is dependent on Cullin-RING E3 ligases (CRLs). Strikingly, neddylation inhibition did not rescue AK59-mediated STING degradation, which suggests that the E3 ligase responsible for AK59-dependent STING degradation is not a CRL. While the proteasomal inhibitors exhibited significant cellular toxicity, bortezomib showed the best rescue of the AK59-mediated STING degradation at two concentrations (25 and 50 nM) without causing

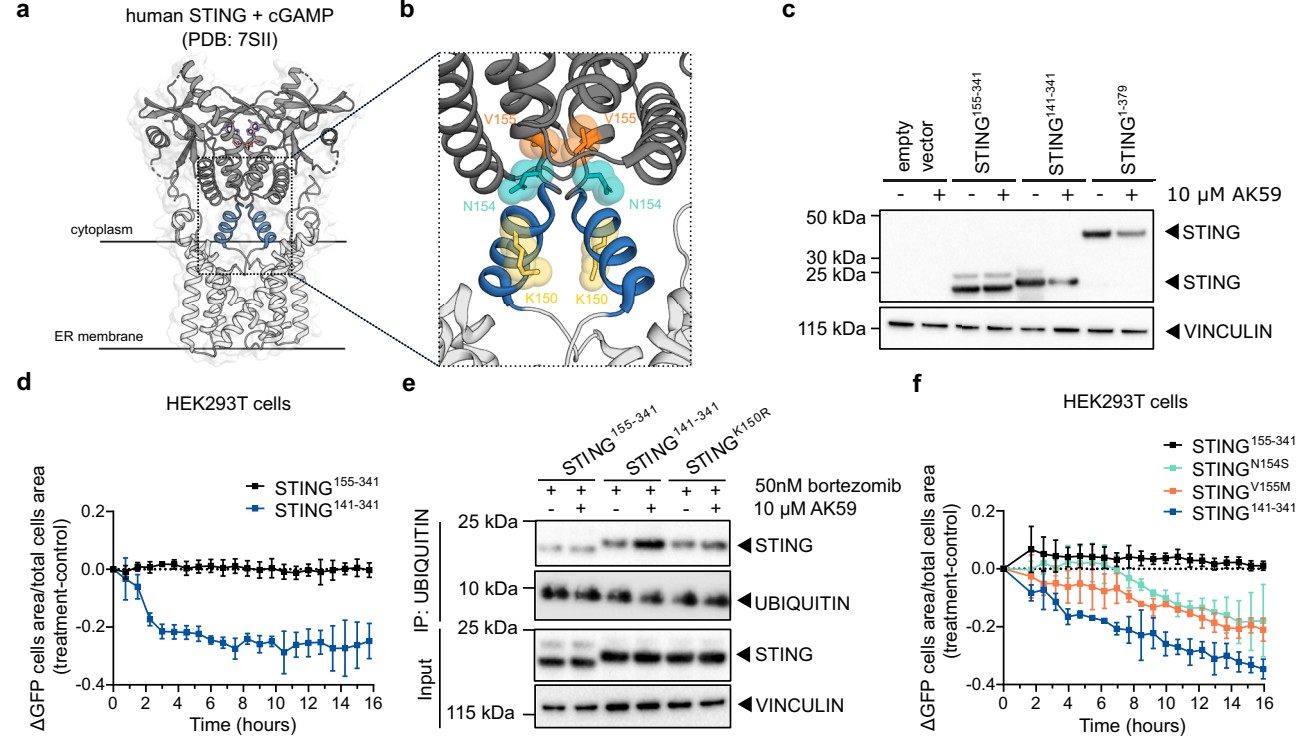

**Fig. 2 | K150 is essential for STING ubiquitination upon AK59 treatment.**
**a** Crystal structure of the dimerized STING[53]. The N-terminus of the structure, containing transmembrane domains are shown in light gray and the C-terminus of the structure containing ligand biding domain is shown in dark gray. The connector helix is shown in blue. cGAMP, represented in pink, bound to the cytosolic domain of STING dimer. **b** STING dimer crystal structure zoomed in to the K150 residue (shown in yellow), N154 residue (shown in cyan), and V155 residue (shown in orange). **c** Western blot showing the STING expression levels in empty vector, STING[155-341], STING[141-341], or STING[1-379] (not GFP tagged) expressing in HEK293T cells. Cells were either treated with 10 μM of AK59 or DMSO control for 16 h. Results are representative of three independent experiments. **d** Live cell tracking of C-terminally GFP-tagged STING[141-341]-GFP or STING[155-341]-GFP. For each data point, 16 pictures/well were averaged per biological replicate. Three biological replicates were plotted as mean ± SD. **e** Ubiquitin pulldown followed by western blot on HEK293T cells transfected with STING[141-341], STING[155-341], STING[141-341_K150R] (not GFP tagged) and treated with either 10 μM AK59 or DMSO (vehicle) control for 16 h. All samples were treated with 50 nM bortezomib 1 h prior to the AK59 (or DMSO control) treatment. Results are representative of three independent experiments. **f** Live cell tracking of C-terminally GFP-tagged STING[141-341]-GFP and SAVI mutants STING[141-341_N154S]-GFP, STING[141-341_V155M]-GFP transfected HEK293T cells after 10 μM AK59 treatment. The connector helix deficient construct STING[155-341] was used as a negative control. GFP+ cell area divided by total cell area normalized to the start of the treatment and delta GFP signal calculated by (AK59 treatment-DMSO control) for each time point. For each data point, 16 pictures/well were taken. Data from biological replicates were plotted as mean ± SD. Source data are provided as a Source Data file.

excessive cell death when combined with various doses of AK59. FACS analysis of STING protein levels in THP1 cells (Fig. 1g and Supplementary Fig. 1f) showed that proteasomal inhibition via bortezomib partially rescued STING degradation at various AK59 doses. Furthermore, ubiquitination pulldown assays showed STING ubiquitination upon AK59 (Fig. 1h). In conclusion, our findings reveal that AK59 inhibits cGAS/STING signaling by inducing STING degradation via the UPS.

## Lysine 150 of STING is essential for AK59-mediated STING degradation

We next aimed to dissect the functional domains in STING which become ubiquitinated upon AK59 treatment and mediate STING degradation. STING contains a ~150 amino acid transmembrane domain with two lysine residues (K20 and K137) in its N-terminus (Supplementary Fig. 2a and Fig. 2a). The C-terminus is comprised of a ~250 amino acid cytosolic ligand-binding domain responsible for cGAMP binding and STING dimerization and contains seven lysine residues (K150, K224, K236, K289, K338, K347, and K370). The ligand-binding domain and the transmembrane domains are connected by a connector helix which contains one lysine residue (K150, Fig. 2a, b). Numerous structural studies have demonstrated the feasibility of expressing the ligand-binding domain of STING in isolation while preserving its structure and capability of cGAMP binding. We thus

chose HEK293T cells where STING is not expressed endogenously and transiently expressed full-length STING (STING[1-379]), the isolated STING ligand-binding domain (STING[155-341]) or the ligand-binding domain including the connector helix (STING[141-341]). As a control, we transfected an empty vector. Forty-eight hours post-transfection, cells were treated with 10 μM AK59 or DMSO control. While all the constructs were expressed in comparable amounts, we observed an AK59-dependent decrease in STING levels only in STING[1-379] and STING[141-341] expressing cells but not in cells expression STING[155-341] (Fig. 2c).

As a complementary approach, we expressed the same constructs with C-terminal GFP-tags in HEK293T cells for live tracking of STING degradation. Interestingly, we observed that STING[1-379]-GFP expressing cells showing aberrant differences in the live tracing (Supplementary Fig. 2b). Considering the similar protein expression levels, we hypothesized that this difference might be due to the slow growth rate of these cells, which might be due to the complexity of transmembrane domain folding. For better comparability in follow up experiments, we therefore proceeded only with expression constructs of the cytosolic domain of STING. To observe the dependency of STING degradation on the connector helix by the live cell tracing experiments, we treated STING[141-341]-GFP or STING[155-341]-GFP expressing HEK293T cells with 10 μM AK59 or DMSO control and monitored the GFP signal as a readout of STING expression levels (Supplementary Fig. 2a). The GFP positive cell area was normalized to the

total cell area for each image in order to compensate for confluency specific variations and samples were normalized to the treatment starting point. The change in the GFP signal (ΔGFP) was calculated by subtracting the fold GFP signal change of the DMSO control group from the signal change in the AK59 treatment group. STING[141-341] showed a clear GFP signal decrease indicating STING degradation upon AK59 treatment. By contrast, STING[155-341]-GFP expressing HEK293T cells showed a stable GFP signal after 10 μM of AK59 treatment (Fig. 2d), indicating stable STING levels. These observations align well with our previous western blot analysis (Fig. 2c) and indicate that the connector helix of STING (residues 141–155) is involved in the function of AK59.

To further understand the role of the connector helix for AK59 function, we focused our efforts on K150 within the connector helix (Fig. 2b) since it has previously been linked to PTMs such as ubiquitination followed by degradation of STING[30,31]. To probe whether K150 is also a ubiquitination site in the mechanism underlying AK59-induced STING degradation, we mutated K150 to arginine (K150R, Supplementary Fig. 2a) in non-GFP tagged STING[141-341] (STING[141-341,K150R]). In a side-by-side comparison with non-GFP-tagged STING[141-341] and STING[155-341], STING[141-341,K150R] was transiently expressed in HEK293T and, 48-h after transfection, cells were treated with 50 nM bortezomib for 1 h prior to the 16-hour AK59 or DMSO control treatment. Since HEK293T cells do not express endogenous STING and transient overexpression causing high peak of expression, we disregarded potential residual ubiquitination of transiently expressed constructs. Ubiquitin pulldown followed by western blot showed an increase in compound-dependent ubiquitination of STING[141-341], whereas ubiquitination of STING[155-341] or STING[141-341,K150R] remained constant (Fig. 2e). These data indicated that residue K150 of STING is essential for the AK59-mediated ubiquitination and therefore degradation of the protein.

Next, we investigated whether AK59 is also capable of inducing the degradation of pathological STING variants. We focused our efforts on SAVI, a rare genetic autoimmune disorder caused by single-point mutations in STING[32–35]. The two most commonly detective causative point mutations in SAVI patients are N154S and V155M in the cytosolic domain of STING[36]. In order to observe the effect of AK59 on SAVI mutant STING proteins, we generated STING[141-341]-GFP constructs carrying the respective mutations (Supplementary Fig. 2a and Fig. 2b). After transient transfection of the wildtype and mutant STING[141-341] constructs, GFP signals were tracked in vehicle treated or 10 μM AK59-treated cells. Tracking of live cells revealed that, similar to STING[141-341]-GFP, both STING SAVI mutants were partially degraded upon AK59 treatment (Fig. 2f and Supplementary Fig. 2c). Our data suggest that AK59 is still able to induce the degradation of STING-SAVI mutants. This finding, together with the observation that AK59 can induce STING degradation in CD14+ human PBMCs highlights AK59 as a potential starting point for the development of SAVI therapeutics.

## AK59 function relies on HERC4, UBA5, and UBA6

While identifying the necessary domains and the lysine residue for AK59-mediated STING degradation, the exact mechanism of action of the compound remained unknown. Therefore, a pooled genome-wide CRISPR-Cas9 knockout screen was conducted to identify genes required for AK59 to reduce STING protein levels in THP1-Cas9 cells[37]. The aim was to identify genes that, when knocked out, significantly abrogate the effect of AK59 in reducing the levels of STING protein in THP1-Cas9 cells.

A FACS-based assay to monitor the levels of STING protein was developed and used as a phenotypic readout for the CRISPR-Cas9 screen (Fig. 3a). Compound concentration and incubation time were optimized according to the median fold change of STING staining (Fig. 3b and Supplementary Fig. 1f). This identified 10 μM AK59 and a 16-h incubation as giving a distinctive 5-to-6-fold separation in the STING protein levels in our FACS assay. After establishing the

FACS-based STING degradation assay, the CRISPR-Cas9 genome-wide screen was conducted. The cells were subjected to either DMSO (vehicle) control or 10 μM of AK59 for 16 h and then sorted according to their STING expression levels. High and low STING expression groups were assigned as the higher and lower 25% quartiles of the sorted cells. Enriched and depleted sgRNAs in the comparison of high vs. low STING protein levels following treatment, as well as the control group, were presented as a plot displaying RSA $p$ value (significance of the hits) for guides targeting each gene and the magnitude of this effect compared to the other genes (Q3 values) (Fig. 3c, Supplementary Fig. 3a–c, and Supplementary Data 2). To distinguish the genes that are significantly enriched in the AK59 high vs. low STING comparison and stable in the DMSO high vs. low comparison, the Euclidean distance (vectoral ranking) between the two sets of (Q, RSA) coordinates was calculated (Supplementary Data 2, 4). Ranking the distances enabled us to identify the hits that are altered only in the AK59 low vs. high comparison but not in the corresponding DMSO comparison (Fig. 3d). Among the highly ranked genes, HECT and RLD domain-containing E3 ubiquitin protein ligase 4 (HERC4), ubiquitin-like modifier activating enzyme 5 (UBA5) and ubiquitin-like modifier activating enzyme 6 (UBA6) attracted our attention. Individual sgRNAs in the screen targeting these three genes show that most of them indeed enriched in the AK59 treatment group with high STING expression compared to the control group (Fig. 3e). Additionally, STRING analysis (Version 11.5)[38] established a network between UBA5, UBA6 and HERC4 in human. Importantly, none of these candidates have been previously linked to STING (Supplementary Fig. 3d). These results increased our confidence that the mechanism underlying the AK59-mediated STING degradation acts through the UPS involving HERC4, UBA5, and UBA6 as regulatory factors.

To validate the CRISPR genome-wide screen results, individual knockout cell lines targeting either *HERC4*, *UBA5*, or *UBA6* were constructed. The two best-performing sgRNAs per gene candidate from the genome library were selected and introduced into THP1-Cas9 cells with lentiviral particles. After the third passaging of the cells following lentiviral transduction, cells were collected to check the CRISPR editing efficiency. The modification rate on the cut-site was assessed by PCR followed by tracking of indels by decomposition (TIDE) analysis[39] (Fig. 4a). To show that the modifications on the cut sites on each gene resulted in depletion (or decrease) of the respective genes protein levels were assessed by western blot analysis (Fig. 4b–d). ACTIN or VINCULIN proteins were used as a loading control. After confirming the knockout of each gene, the FACS-based assay monitoring STING protein levels was used to further validate the screening results. *HERC4*, *UBA5*, and *UBA6* sgRNA transduced THP1-Cas9 lines showed higher levels of STING protein in the presence of 10 μM AK59 treatment compared to control (Ctrl) sgRNA transduced THP1-Cas9 cells (Fig. 4e and Supplementary Figs. 1f, 4). While AK59 treatment resulted in up to a 70% decrease of STING protein in the control cell line, STING depletion was only 40–50% in the *HERC4, UBA5*, and *UBA6* knockout cell lines. Furthermore, *HERC4*, as the second top-rated hit in the CRISPR screen, showed the highest rescue phenotype regarding STING expression in the AK59 treatment group (Fig. 4f). All these data indicate that we could replicate the results of the genome-wide screen with individual knockout lines and that loss of *HERC4, UBA5*, and *UBA6* perturbs the effect of AK59 on the reduction of STING protein levels.

## AK59 degrades STING by inducing its interaction with the E3 ligase HERC4

Among the validated top hits of the CRISRP-Cas9 screen, UBA5 and UBA6 are E1 enzymes suggesting they may act as accessory elements in the mechanism underlying the AK59-mediated STING degradation via UPS. HERC4, in contrast, is a HECT-type E3 ligase and might be directly responsible for the ubiquitination of STING followed by its proteasomal degradation in the presence of AK59. We, therefore, prioritized

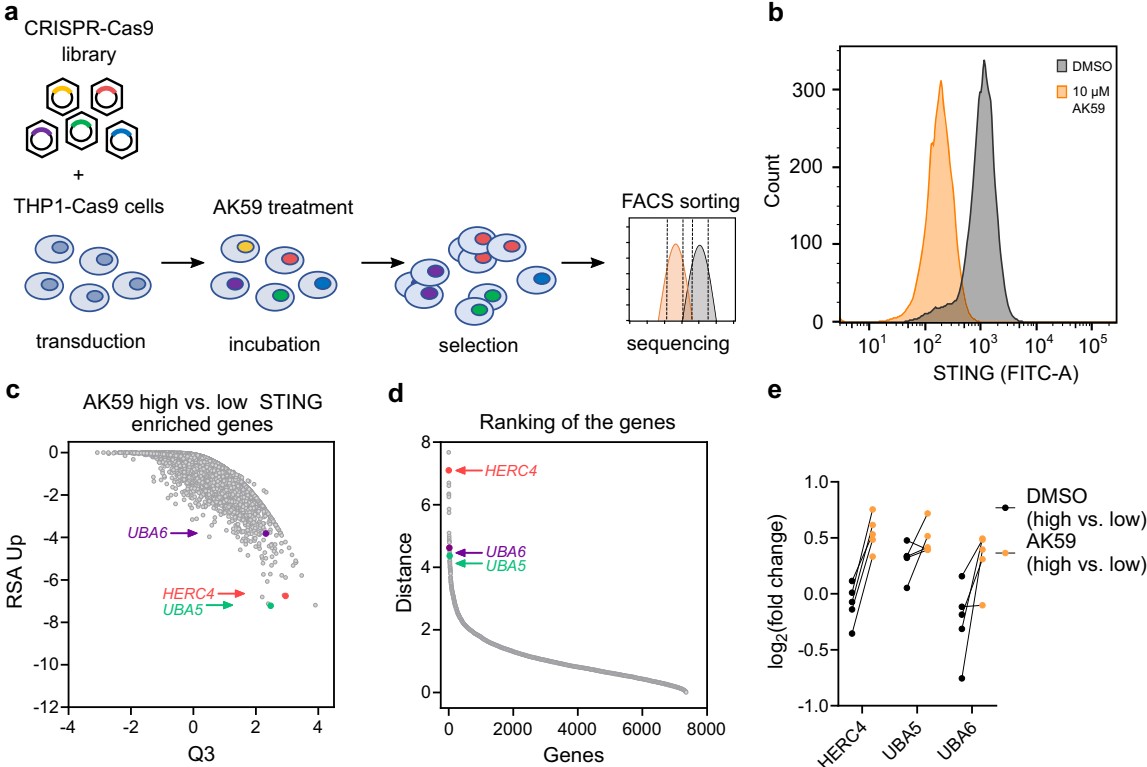

**Fig. 3 | CRISPR-Cas9 genome-wide screening unveils genes responsible for AK59 activity on STING expression. a** Schematic representation of the workflow of CRISPR-Cas9 genome-wide knockout screen. See the main text or Materials and methods section for details. **b** FACS analysis of THP1-Cas9 cells treated with either DMSO (black) or 10 μM AK59 (orange). Detection of STING expression was repeated in three biological replicates and representative FACS reads were plotted using FlowJo (Version 10.6.1). **c** Comparison of RSA up values with Q3 values from the CRISPR-Cas9 screen of 10 μM treated THP1 cells represented in a dot plot. The data represented in the plot corresponds to the comparison between high STING expression versus low STING expression in the AK59 treatment group. Each dot represents a gene from the CRISPR-Cas9 library. **d** Ranking of genes that upon AK59 treatment of THP1 cells, have enriched sgRNAs in the high STING expression group compared to the low expression group. The ranking distance was calculated as the change of RSA and Q values between the two treatment groups. Each gene is represented with a dot on the graph. **e** Individual fold changes of each sgRNA targeting *HERC4*, *UBA5*, and *UBA6* in the high vs. low STING expression comparison groups in the compound treated and untreated samples. Source data are provided as a Source Data file.

HERC4 for further analysis with the hypothesis that AK59 is acting by establishing a distinct interaction between HERC4 and STING, allowing STING to be ubiquitinated and degraded. To test this hypothesis, we first validated that not only compound-mediated STING degradation, but also cGAS/STING pathway inhibition was ablated by *HERC4* knockout. To link HERC4 to the function of AK59, *HERC4* sgRNA was introduced to the Dual-THP1-Cas9[40] cell line to knockout *HERC4*. Validation of knockout was performed both by TIDE analysis on the cut-site as well as western blot analysis on the protein level (Supplementary Fig. 5a, b). As expected, luminescence due to cGAMP stimulation, indicating activation of the IRF pathway, gradually decreased with increasing concentrations of AK59 in the control cell line (Fig. 5a). By contrast, the *HERC4* knockout Dual-THP1-Cas9 cell line showed higher luciferase activity in the presence of AK59 indicating that the compound's inhibitory activity on the IRF pathway is partially compromised in the absence of HERC4. To support this data, we analyzed phosphorylation levels of IRF3 and TBK1 as well as CXCL10 and IFN-beta release upon cGAMP stimulation either in control or HERC4 sgRNA transduced Dual-THP1 cells. Western blotting showed decreasing levels of IRF3 and TBK1 phosphorylation upon AK59 treatment compared to QK50 treated samples (Supplementary Fig. 5c). Interestingly, we also observed that HERC4 protein levels decreased upon AK59 treatment. In parallel to phosphorylation data, CXCL10 and IFNβ amounts shows significant increasing trend in *HERC4* knockout cells compared to control group (Supplementary Fig. 5d, e). Thus, the absence of HERC4 interrupted the effect of AK59 on STING

protein levels and downstream on activation of the IRF pathway. However, it is important to point out that in all assays investigating downstream STING pathway activity (IRF reporter assay, phosphorylation of IRF3 and TBK1, CXCL10, and IFNβ levels), *HERC4* knockout lines still show downstream pathway inhibition with increasing doses of AK59, meaning that HERC4 does not fully rescue AK59-dependent cGAS/STING pathway inhibition.

To further investigate the link between HERC4 and the degradation of STING through AK59, *HERC4* was knocked out in HEK293-JumpIN-Cas9 cells, and knockout efficiency was shown both by TIDE analysis and western blot (Supplementary Fig. 5f, g). Subsequently, GFP-fused constructs compromising the cytosolic domain for STING were transiently expressed in the generated knockout line (Supplementary Fig. 2a). GFP signal tracking in living *HERC4* knockout cells that were transfected with STING-GFP constructs showed that *HERC4* knockout ablates the differences in the degradation between different STING constructs (Fig. 5b). In other words, degradation of cytosolic STING via AK59 was partially blocked in the absence of HERC4.

At this point, we wanted to investigate how AK59 modulates the substrate specificity of HERC4 beyond STING. Therefore, we analyzed the potential changes in the protein levels of known substrates of HERC4, such as SMO and PGR[24,25,28] in the presence of AK59 (Supplementary Fig. 6a). Both in wildtype Dual-THP1-Cas9 cells as well as in *STING* KO cells, protein levels of SMO, as well as PGR, remained unchanged or even stabilized upon treatment suggesting that AK59 does not function on degradation of known native HERC4 substrates.

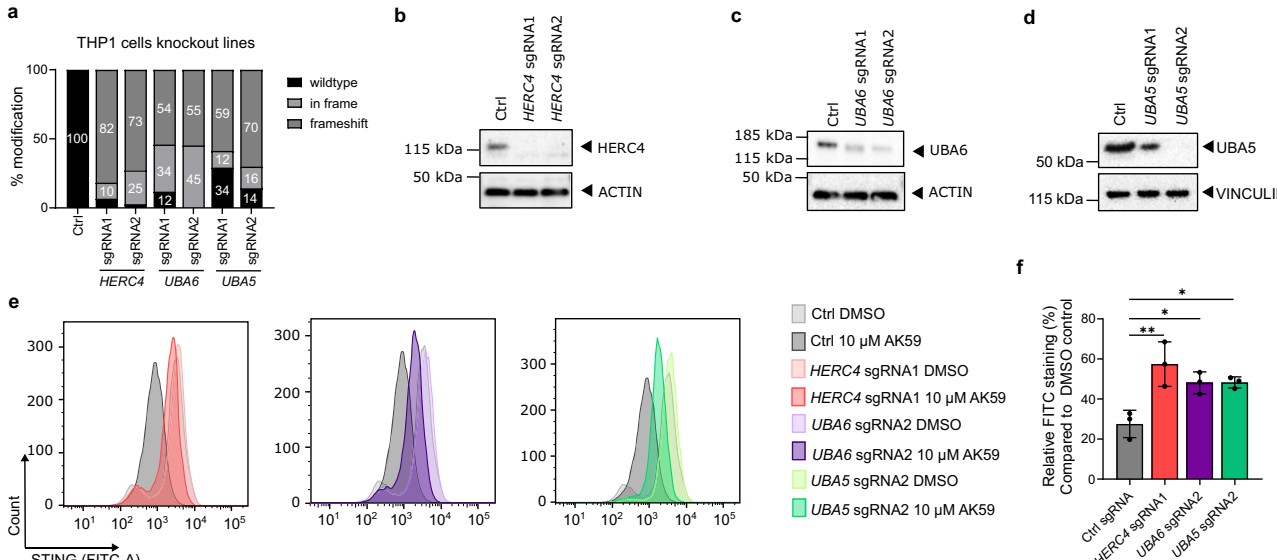

**Fig. 4 | Validation of *HERC4, UBA5*, and *UBA6* as genes responsible for AK59 activity on STING levels. a** TIDE analysis of either Ctrl sgRNA, sgRNA1 or sgRNA2 transduced THP1-Cas9 cells for *HERC4, UBA5*, and *UBA6*. Modification rates were checked at the third passage after the initial transduction. Undefined sequencing reads were excluded. **b**–**d** Western blot to detect HERC4 (**b**), UBA5 (**c**), and UBA6 (**d**) protein levels of either Ctrl sgRNA, sgRNA1, or sgRNA2 transduced THP1 cells. Proteins from each cell line were collected at the third passage after the initial transduction. ACTIN or VINCULIN was used as a loading control. Results are representative of 3 independent experiments. **e** FACS analysis of STING expression on Ctrl sgRNA or *HERC4* sgRNA1, *UBA6* sgRNA2, *UBA5* sgRNA2 transduced THP1-

Cas9 cells that were treated with either DMSO or 10 μM AK59. The experiment was repeated in three biological replicates and representative FACS reads plotted using FlowJo (Version 10.6.1). **f** Relative STING expression levels *HERC4* sgRNA1, *UBA5* sgRNA2, and *UBA6* sgRNA2 transduced THP1-Cas9 cell lines determined via FACS analysis. Data were normalized to the DMSO (vehicle) treated group and three biological replicates were plotted as mean ± SD. Statistical significance was calculated using one-way ANOVA followed by Dunnett's multiple comparison test. Significance indicated as **$p = 0.0024$, *$p = 0.0202$, and *$p = 0.0194$ respectively. Source data are provided as a Source Data file.

Furthermore, we performed proteomics analysis on *HERC4* knockout or *STING* knockout Dual-THP1 cells to investigate compound-dependent protein expression changes in an unbiased manner (Supplementary Fig. 6b, c and Supplementary Data 3). As expected, STING expression remained unchanged in AK59 versus control-treated *HERC4* knockout Dual-THP1 cells, further indicating the HERC4-dependency of AK59-derived STING degradation (Supplementary Fig. 6b). Unfortunately, due to peptide detection limitations, we could not detect SMO or PGR in our proteomic analysis results to further support the western blot results. Interestingly, the proteomics data on AK59-treated *HERC4* KO cells, as well as *STING* KO cells, revealed multiple candidates that are downregulated by AK59 independent of HERC4 or STING function (Supplementary Fig. 6b, c and Supplementary Data 3). These findings show that, while it is important to acknowledge STING degradation via HERC4, secondary or indirect functions of AK59 should be taken into consideration.

Interestingly, while we were primarily focusing on protein level changes of HERC4 targets, we also observed that HERC4 levels decrease in an AK59-dependent manner (Supplementary Fig. 6a, lane 2 and 4). In fact, HERC4 was also one of the proteins downregulated upon AK59 treatment in our initial proteomics data (Fig. 1e, f, Supplementary Data 1, HERC4 LFC: −0.72, and *q* value: 1.63810-8 in AK59 vs. DMSO comparison). We hypothesized that these complementing results reflect a compound-mediated disruption of an intramolecular autoinhibitory state of HERC4 which might lead to its degradation[41,42]. However, this hypothesis requires further investigation and a deeper understanding of HERC4 function.

Investigation of known HERC4 substrates, as well as knock-out proteomic data, brought us one step closer to the hypothesis that AK59 acts as a STING degrader through the E3 ligase HERC4 and suggested that AK59 could be functioning as an MGD that redirects HERC4 to neosubstrate STING. The definition of an MGD implies that it induces a physical interaction between an E3 ligase and a neosubstrate.

In order to validate whether AK59 acts as a HERC4-dependent STING degrader, we probed for a potential compound-dependent PPI between HERC4 and STING, first by co-immunoprecipitation followed by western blot using transiently STING[141-341] expressing HEK293T cells. After 48 h post-transfection with either empty vector or STING[141-341] construct, HEK293T cells were treated with DMSO or AK59. Co-immunoprecipitation, followed by western blot analysis showed that only in the presence of AK59, STING was co-immunoprecipitated together with HERC4 (Fig. 5c, third lane). Importantly, STING did not co-immunoprecipitate with HERC4 when the cells were only treated with DMSO (Fig. 5c, second lane).

To support the pulldown data indicating the AK59-mediated interaction between HERC4 and STING, we utilized a NanoBiT® complementation assay (Promega). The NanoBiT® complementation assay consists of a small (SmBiT) and large (LgBiT) complementary domain of the Nanoluciferase protein, which have low binding affinity for each other. Only in the presence of direct interaction between the conjugated protein domains, the SmBiT and LgBiT form a functional Nanoluc®(Nluc) and produce a luminescence signal[43]. Complex formation between the conjugated proteins can, therefore, be detected as an increase in luminescence. As complementary expression constructs, we conjugated the STING[141-341] cytosolic domain C-terminally to the LgBiT domain and full-length HERC4, N-terminally to the SmBiT moiety (Supplementary Fig. 7a). Both constructs were then transiently transfected into HEK293T cells and expression of the constructs was validated via western blot (Supplementary Fig. 7a, below). Then 48 h post-transfection, cells were treated with either DMSO, AK59, or QK50. Time course (Fig. 5d), as well as dose-dependent (Fig. 5e) measurements, showed a significant luminescence increase after 8 h in the AK59-treated cells while the luminescence intensity remained at basal levels in the DMSO and QK50 treated cells. This observation indicates that only AK59, but not the inactive analog QK50, is capable of inducing an interaction

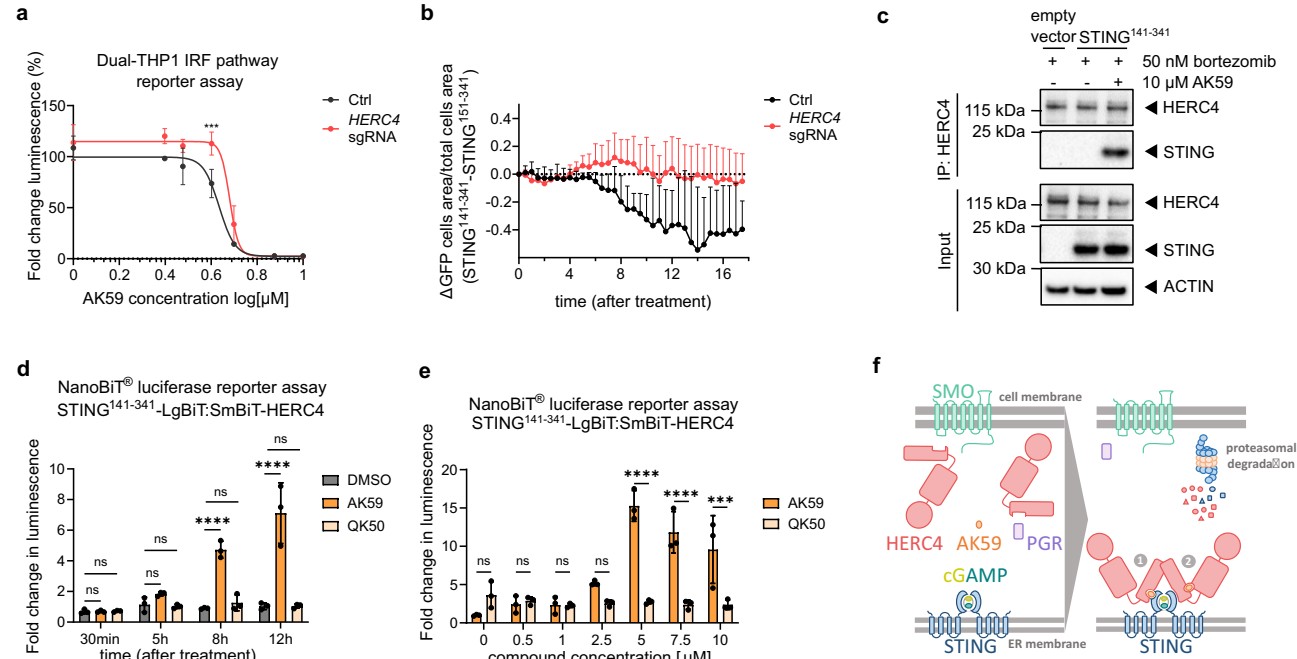

**Fig. 5 | AK59 acts as STING degrader mediated by HERC4. a** IRF pathway reporter assay on wildtype or *HERC4* knockout Dual-THP1-Cas9 cells. Data plotted as mean ± SD of three individual biological replicates. Calculated half-maximal inhibitory concentrations (IC50s) are 4.316 and 4.792 for Ctrl and *HERC4* sgRNA, respectively. Significance was calculated using two-way ANOVA followed by Šidák's correction. Significance indicated as ***$p < 0.001$. **b** Live cell tracking of STING$^{141-341}$-GFP and STING$^{151-341}$-GFP in wildtype or *HERC4* knockout HEK293-JumpIN-Cas9 cells after 10 μM AK59 treatment. Four biological replicates were plotted as mean ± SD. **c** Co-immunoprecipitation with HERC4 followed by western blot of indicated proteins on HEK293T cells transiently expressing STING$^{141-341}$ and treated either with 10 μM AK59 or DMSO control. Transfection of an empty vector was used as a negative control. Results are representative of 3 independent experiments. **d** NanoBiT® assay on STING-LgBiT and SmBiT-HERC4 expressing HEK293T cells treated with either 10 μM AK59, 10 μM QK50, or DMSO control. Three biological replicates were plotted as mean ± SD. Significance was calculated using two-way ANOVA followed by Šidák's correction. Significance indicated as ns $p > 0.05$, ****$p < 0.0001$. **e** NanoBiT® assay on STING-LgBiT and SmBiT-HERC4 expressing HEK293T cells in the presence of increasing concentrations of either AK59 or QK50 over 16-hour treatment. Three biological replicates were plotted as mean ± SD. Significance was calculated using two-way ANOVA followed by Šidák's correction. Significance indicated as ns $p > 0.05$, ***$p = 0.0007$, and ****$p < 0.0001$. **f** Graphical representation of naive and compound added state of the cells and the suggested protein–protein interactions in the presence of AK59. The panel represents two possible PPI induced by AK59: with AK59 interacting directly with both STING and HERC4 (indicated as 1) or via an allosteric site on HERC4, allowing neosubstrate recognition (indicated as 2). Source data are provided as a Source Data file.

between SmBiT-HERC4 and STING$^{141-341}$-LgBiT that is compatible with the formation of an active Nluc. Additionally, we wanted to investigate whether the quinoline moiety of AK59 has a binary binding affinity to either HERC4 or STING that might affect complex formation between HERC4 and STING. In order to test this, we leveraged the QK50 by competing with AK59 in the Nanoluc reporter assay. STING$^{141-341}$LgBiT: SmBiT-HERC4 expressing cells were treated with 10 μM QK50 prior to the increased dose of AK59 treatment. We hypothesized if QK50 had a binary affinity to either HERC4 or STING, we would observe a decrease in the fold change of luminescence upon AK59 addition when cells are preincubated with QK50. As a control, we used equivalent DMSO treatment prior to the addition of AK59 (Supplementary Fig. 7b). We observed a similar increase in luminescence over the tested range of AK59 doses between cells preincubated with DMSO as control and QK50, indicating that QK50 does not compete with AK59 induces HERC4-STING complex formation. Thus, QK50 does not seem to have a significant binary affinity for either HERC4 or STING which highlights the steep SAR regulating this compound-protein interaction. Finally, we tested whether cGAMP, the natural agonist of STING, competes with the AK59-induced STING-HERC4 in our NanoBiT® assay by supplementing cells with 30 μM cGAMP prior to incubation with AK59 (Supplementary Fig. 7c). We did not observe any reduction in the Nluc signal in the cGAMP treated cells compared to untreated cells. This suggests that AK59, unlike cGAMP, does not bind to the active site in STING. Together, the co-immunoprecipitation and NanoBiT® assay data demonstrate the ability of AK59 for inducing a PPI between

HERC4 and STING, potentially through a ternary complex involving the compound and both protein interaction partners.

HECT-domain E3 ligases contain an active cysteine site for direct ubiquitination of HERC4 before transfer of the ubiquitin modification to the target[44]. In order to stabilize the complex formation, the predicted active cysteine site on HERC4 was mutated to alanine in the NanoBiT® pair construct. The hypothesis is to block the ubiquitin-binding to the HERC4 and prolong the luminescence signal before the degradation of STING. While both wildtype as well as active cysteine mutant HERC4 proteins bind to STING$^{141-341}$ protein in the presence of increasing doses of AK59, HERC4$^{C1025A}$ shows slightly higher fold change in the luminescence reads compared to wild-type HERC4 (Supplementary Fig. 7d).

While we demonstrated that K150 in the connector helix is the site important for STING ubiquitination (Fig. 2), we further investigated which parts of STING are essential for complex formation with HERC4 induced by AK59. To test this, we transiently expressed STING$^{141-341}$, STING$^{155-341}$, and STING$^{141-341\ K150R}$ in HEK293T cells and probed their interaction with HERC4 after AK59 treatment via co-immunoprecipitation. The co-immunoprecipitation was performed on 50 nM bortezomib-treated cells prior to 16-hour DMSO control or AK59 treatment to prevent STING and HERC4 degradation. Interestingly, all tested STING constructs were found to still be interacting with the tested STING constructs upon compound incubation, as shown by co-immunoprecipitation (Supplementary Fig. 7e). This demonstrates that the connector helix and residue K150 in STING, while being essential for ubiquitination, are not necessary for AK59 induced

complex formation with HERC4. This finding was supported by the NanoBiT® complementation assay, where various STING constructs were cloned in the LgBiT-tagging plasmid pair. Each STING construct (STING[155-341], STING[141-341], and STING[141-341_K150R]) co-transfected with SmBiT-HERC4 expression construct and AK59-driven complex formation was tested by luminescence read out. While all constructs showed a luminescence signal indicating a direct interaction with HERC4 upon AK59, we did not observe any significant difference between constructs (Supplementary Fig. 7f). This corroborates our findings from co-immunoprecipitation studies that the connector helix as well as residue K150 are not necessary for the AK59-induced interaction of STING with HERC4.

Overall, our findings describe the discovery of AK59 as a STING degrader that acts by inducing an interaction between STING and the HECT-type E3 ligase, HERC4. AK59 shows several characteristics of an MGD, for example, it redirects STING as a neosubstrate to HERC4. This induced PPI between STING and HERC4 results in the ubiquitination of STING at K150 followed by its proteasomal degradation (Fig. 5f). Importantly, STING degradation is translated downstream into IRF pathway inhibition. Additionally, we showed that AK59 not only degrades wild-type STING in close-to clinic human CD14+ monocytic cells but also SAVI mutant STING proteins and therefore might present a starting point for the development antagonists to treat STING related diseases.

## Discussion

In this study, we report the identification of AK59 as a small molecule STING degrader that efficiently inhibits the cGAS/STING pathway. Targeting STING has been of high interest for drug discovery due to its regulatory role in interferon signaling. This report demonstrates that AK59 causes degradation of STING, thus inhibiting downstream phosphorylation of elements of the cGAS/STING pathway, such as IRF3 and TBK1, and subsequently reducing the release of cytokines such as IFNβ and CXCL10.

We leveraged unbiased cellular proteomics to dissect the function of the small molecule AK59 and could show that STING is not only downregulated upon AK59 treatment, but also that STING gets degraded through the ubiquitin-proteasome system. Importantly, STING was not the only downregulated protein upon AK59 treatment. The protein levels of multiple proteins associated with cellular stress were differentially altered. This effect was recapitulated in STING CRISPR/Cas9 knockout cell lines treated with AK59, suggesting that the AK59-dependent proteome-wide changes are not an effect of STING degradation but rather AK59 off-target effects.

We narrowed down the STING domains necessary for the AK59 function with a divide-and-conquer approach. By tracking GFP-tagged STING deletion constructs in live cells as well as end-point protein level measurements, we could show that the cytosolic domain of STING is essential for AK59-dependent degradation. The same domain is responsible for the binding of the physiological STING agonist cGAMP. Using single-point mutations, we dissected K150 in the connector helix as the potential ubiquitination site of STING responsible for AK59-dependent STING degradation. Ubiquitination of the same residue was previously implicated in proteasomal STING degradation after viral infection[30,45]. Importantly, AK59 was capable of inducing STING degradation in human CD14+ monocytes and showed activity on STING proteins carrying SAVI mutations. There have been numerous attempts to inhibit the cGAS/STING pathway by targeting STING either via small molecule inhibitors, cGAMP derivatives, or with PROTACs[11,46,47]. To our knowledge, inhibitors or degraders of STING SAVI mutants other than AK59 have not been described so far.

A CRISPR/Cas9 genome-wide knockout screen allowed us to identify key regulators as well as accessory factors of the AK59-dependent STING degradation mechanism. We focused our efforts on the screening hit *HERC4*, a HECT- E3 ligase. Individual CRISPR/Cas9

mediated knockout of HERC4 reduced the effect of AK59 on STING degradation and cGAS/STING pathway inhibition. We thus hypothesized that AK59 may act as a STING degrader by inducing its ubiquitination through HERC4. Notably, the loss of HERC4 did not fully abrogate the AK59-induced degradation of STING, potentially indicating the existence of compensatory or alternative mechanisms that lead to STING degradation. Besides HERC4, we identified the ubiquitin-activating enzymes UBA5 and UBA6 in our CISRPR/Cas9 screen as co-dependencies for the function of AK59 as a STING degrader. As for HERC4, individual knockout of *UBA5* or *UBA6* only partially blocked STING degradation by AK59. The details of how UBA5, UBA6, and HERC4 act together to induce STING degradation by AK59 remains to be determined.

We demonstrated that AK59 induces a complex with STING and HERC4 in cells using orthogonal co-immunoprecipitation and NanoBiT® complementation assays. Strikingly, the non-functional analog QK50 was not capable of inducing STING-HERC4 complex formation and did not compete with AK59 to prevent complex formation. This highlights the very constrained structure-activity relationship of AK59 for inducing HERC4-STING interactions and potential challenges for future compound optimization efforts. We could also narrow down the HERC4 binding site to the cytosolic domain of STING. Interestingly, K150 and the connector helix were not required for AK59-induced association of HERC4, which suggests a spatial separation between the degron and the ubiquitination site in STING. We did not observe any competition of cGAMP with the AK59-induced complex formation between HERC4 and STING which suggests that AK59 acts independent of the activation status of STING and likely not through the cGAMP binding site. The exact degron sequence in STING recognized by HERC4 and AK59, as well as the binding site of AK59 in HERC4, remains to be determined.

Targeted protein degradation is an emerging modality in drug discovery in which E3 ligases are rewired by small molecules to recognize neosubstrates. While most E3 ligases exploited for TPD efforts are part of the RING E3 ligase family, the HECT domain-containing E3 ligases such as HERC4 are yet to emerge on the TPD landscape. A crucial feature of HECT-type E3 ligases is a catalytic cysteine in the HECT domain which forms a thioester intermediate with ubiquitin before transferring ubiquitin to the substrate. This is in stark contrast to the scaffolding mechanism of RING ligases, which induce proximity between a substrate and a ubiquitin-conjugating enzyme. Multiple HECT ligases (e.g., HACE1 and NEDD4) have been described to be regulated by an autoinhibitory interaction of N-terminal domains, usually substrate binding sites, with the catalytic HECT domain[41,48]. This autoinhibition can occur through intramolecular interactions or via dimerization and prevents autoubiquitination and degradation of HECT ligases. It is released upon binding of substrates, followed by substrate ubiquitination and autoubiquitination as a side reaction. Likewise, HECT ligase autoubiquitination can be induced by disrupting the autoinhibiting interactions via mutagenesis. Whether HERC4 is also regulated through an autoinhibited state is yet unclear.

We observed an AK59-dependent and proteasome-dependent depletion of HERC4 protein levels in our experiments. This effect was preserved in STING knockout cells, which suggests that AK59-mediated HERC4 depletion is independent of a substrate. These observations strongly suggest that HERC4, indeed is regulated by an autoinhibition mechanism like other HECT ligases. AK59 may release HERC4 from such an autoinhibited state, thereby inducing its auto-ubiquitination and degradation along with ubiquitination and degradation of STING. Consistent with these findings, two scenarios are conceivable for the mode of action of AK59. First, AK59 might function as a classical orthosteric glue that mediates ternary complex formation directly by sitting at the binding interface between HERC4 and STING. Second, AK59 might be an allosteric glue, i.e., by binding to a distal

pocket it either HERC4 or STING, it induces binding surfaces complementary for the respective binding partner (see Fig. 5f). Follow-up biophysical ternary complex formation studies using recombinant proteins as well as high resolution structure elucidation of the ternary complex comprising STING, HERC4 and AK59 will be indispensable for testing these models.

AK59 acts as a degrader of STING functioning through HERC4 in an MGD-like manner. MGDs and bi-functional PROTACS have attracted significant interest as tools or drugs that enable TPD, an event-driven pharmacological modality. Their catalytic activity is rooted in their ability to bring about an interaction between an E3 ligase and a neo-substrate which is subsequently ubiquitinated and degraded via the proteasome[49]. Recently, Liu et al. showcased how a known small molecule inhibitor of STING (C-170) can be repurposed, in a PROTAC approach, for STING degradation by connecting it to the CRBN ligand pomalidomide via a short linker[47]. While rationally designing PROTACs from known target protein inhibitors or binders can be straightforward, the discovery of molecular glues is still driven largely by serendipity. However, MGDs have significant advantages over PROTACs due to their size and, therefore, possibly better pharmacodynamic properties. Moreover, the emergence of resistance mechanisms negating the effectiveness of CRBN-based TPD approaches in the clinics, e.g., through downregulation of CRBN expression or genetic mutations in the CRBN gene, have highlighted the need for finding E3 ligases beyond CRBN suitable for TPD[50]. While we acknowledge that AK59 requires further optimization to increase potency and specificity; this report serves as a starting point for TPD targeting STING.

In summary, the results reported here introduce a research area in targeting the cGAS/STING pathway and opens horizons for the regulation of STING in the context of autoimmune disorders.

## Methods

### Donor samples
Anonymized whole blood samples were collected under informed consent from healthy volunteers through the Interregionale Blutspende (IRB) of the Swiss Red Cross (SRK) in Bern, Switzerland. Sample collection was approved by Ethikkommission Nordwest- und Zentralschweiz (EKNZ), approval number Req-2017-00050. Sex, gender, race, and population characteristics are not defined as the samples were anonymized and the research does not fall outside of Swiss Human Research Act. No ethics oversight is needed according to Swiss Law for this study.

### Cell culture
THP1 cells were cultured using RPMI 1640 Medium, GlutaMAX™ Supplement medium (Thermo Fisher Scientific, Cat#61870010) supplemented with 10% heat inactivated-FCS (Bioconcept, Cat#2-01F36-I), 25 mM Hepes (Thermo Fisher Scientific, Cat#15630080), 1 mM sodium pyruvate (Thermo Fisher Scientific, Cat#11360070), and 50 U/ml penicillin-streptomycin (Thermo Fisher Scientific, Cat#15140122). Cells were passaged every 4-5 days by dilution to 0.2-0.3 million cells/ml.

Dual-THP1™ cells (Invivogen, Cat#thpd-nfis) were cultured using RPMI 1640 Medium, GlutaMAX™ Supplement medium (Thermo Fisher Scientific, Cat#61870010) supplemented with 10% heat inactivated-FCS (Bioconcept, Cat#2-01F36-I), 2 mM L-glutamine (Thermo Fisher Scientific, Cat#25030081), 25 mM Hepes (Thermo Fisher Scientific, Cat#15630080), 1 mM sodium pyruvate (Thermo Fisher Scientific, Cat#11360070), and 50 U/ml penicillin-streptomycin (Thermo Fisher Scientific, Cat#15140122). Cells were passaged every 4-5 days by dilution to 0.2-0.3 million cells/ml.

HEK293T or HEK293-JumpIN cells were cultured with DMEM, high glucose, GlutaMAX™ Supplement, pyruvate (Thermo Fisher Scientific, Cat#31966021) supplemented with 10% heat inactivated-FCS (Bioconcept, Cat#2-01F36-I), 10 mM Hepes (Thermo Fisher Scientific, Cat#15630080), and 50 U/ml penicillin-streptomycin (Thermo Fisher

Scientific, Cat#15140122). Cells were passaged every 3–4 days, by diluting the cells 1 to 5, after releasing the attached cells using TrypLE (Thermo Fisher Scientific, Cat#12605010).

For transfection studies, the TransIT-LT1 transfection reagent (Mirus, Cat#MIR2304) was used according to the manufacturer's protocol. For the six-well format (TPP, Cat#Z707759), 1 million HEK293T cells were seeded one day before transfection. For each well, 2.5 µg of DNA in 7.5 µl of TransIT-LT1 transfection reagent diluted in 250 µl optiMEM (Thermo Fisher Scientific, Cat#31985062). After 15 min of incubation of the transfection reagent with DNA, the transfection mix was added dropwise to the cells. Then 48 h after transfection, cells were treated with the compound of interest.

For lentiviral particle production, $1 \times 10^7$ HEK293T cells were seeded on collagen-coated T75 flasks (Corning, Cat#356485) and then 24 h later, the cells were transfected with 1.84 µg of DNA of interest with 2.24 µg of ready-to-use lentiviral packaging plasmid mix (Cellecta, Cat#CPCP-K2A) using TransIT-LT1 transfection reagent (Mirus, Cat#MIR2304) according to the manufacturer's protocol. Subsequently, the media was changed 24 h after transfection and lentivirus-containing supernatants were collected after 72 h. Finally, lentiviral particles were filtered using a 0.45 µm filter (Sartorius, Cat#16537), before concentrating 10-fold using the Lenti-X concentrator (Takara, Cat#631232) according to the manufacturer's protocol.

### Cell viability assay
Compound toxicity was measured by luminescence-based cell viability assay, CellTiter-Glo® Assay (Promega, Cat#G7570) was used according to the manufacturer's instructions. Viability assays were performed in clear bottom 96-well plates (TPP, Cat#ZZ707902) where THP1-Cas9 cell cells were seeded at a cell density of 0.4 million cells/well and with the indicated doses of compounds. Cells were treated right after seeding. Luciferase measurement was performed after an indicated period (0–72 h) of compound incubation. To detect luciferase activity, CellTiter-Glo® Assay (Promega, Cat#G7570) was used according to the manufacturer's protocol with minor changes. The reagent (50 µl) was added on top of the 50 µl cell-compound containing wells in a black with clear bottom 96-well plates (Greiner, Cat#655090) where the bottom of the plate was sealed (PerkinElmer, Cat#6005199). Luminescence was measured for 0.1 s with the EnVision® Multimode plate reader (Revvity, workstation version 1.14.3049.1193). Each of the experiments was repeated in three independent biological replicates. Fold increase in luminescence was calculated by dividing each read by its matching control group (DMSO treated), and results were plotted using GraphPad Prism 9.

### Proteomics
THP1-Cas9, Dual-THP1-Cas9, Dual-THP1-Cas9 STING KO cells or Dual-THP1-Cas9 HERC4 KO cells ($2 \times 10^6$ per well in a six-well plate) were seeded and treated on the same day either with 0.1% (final) DMSO (Sigma-Aldrich, Cat#D8418) or 50 nM bortezomib (Sigma-Aldrich, Cat#504314, diluted in 0.1% DMSO final). Two to three biological replicates were used for proteomics. One hour after bortezomib treatments, 10 µM AK59 (in-house production, see Supplementary Methods) or 10 µM QK50 (in-house production, see Supplementary Methods) (0.1% (DMSO final) was added on top of the cells and incubated for 16 h. DMSO treatment was used as a control. Cells were collected and washed with 1x DPBS (Thermo Fisher Scientific, Cat#14190094) after compound incubation. Before proteomics analysis of the samples, the conditions were checked using western blot (for more details, see western blot) to confirm targeting effects on STING. PXD040291 dataset was run as 16 samples and PXD046677 run as 18 samples.

TMT-labeled peptides were generated with the iST-NHS kit (PreOmics, Cat#P.O.00030) and TMT16plex or TMT18plex reagent (Thermo Fisher Scientific, Cat#A44522, Cat#A52047) using around

$2 \times 10^6$ cells per sample. Equal amounts of labeled peptides were pooled and ~300 µg of pooled peptides were separated on a Waters XBridge BEH C18, 130 Å, 3.5 µm, 150 × 1 mm column with a gradient from 100% buffer A (10 mM ammonium formate in water, pH 11) to 55% buffer A and 45% buffer B (10% (v/v) 10 mM ammonium formate, pH 11 in water and 90% (v/v) acetonitrile) in 60 min with a flow rate of 60 µl/min. Alternating rows of the resulting 72 fractions were pooled into 24 samples, dried, and resuspended in water containing 0.1% formic acid.

The LC-MS analysis was carried out on an EASY-nLC 1200 system coupled to an Orbitrap Fusion Lumos Tribrid mass spectrometer (Thermo Fisher Scientific). Around 1.5 µg of peptides from each sample (each condition in 2–3 biological replicates) were separated on a 25 cm long Aurora Series UHPLC column (Ion Opticks, Cat# AUR3-25075C18) with 75 µm inner diameter and a gradient from 95% buffer A (0.1% formic acid in water) and 5% buffer B (0.1% formic acid in 80% (v/v) acetonitrile and 20% (v/v) water) to 65% buffer A and 35% buffer B in 168 min and then in 9 min to 35% buffer A and 65% buffer B with a flow rate of 400 nl/min. MS1 spectra were acquired at 120k resolution in the Orbitrap, MS2 spectra were acquired after CID activation in the ion trap, and MS3 spectra were acquired after HCD activation with a synchronous precursor selection approach using 5 or 8 notches and 50 K resolution in the Orbitrap. LC-MS raw files were analyzed with Proteome Discoverer 2.4 (Thermo Fisher Scientific).

Briefly, spectra were searched with Sequest HT against the *Homo sapiens* UniProt protein database and common contaminants (2019, 21,494 entries). The database search criteria included 10 ppm precursor mass tolerance, 0.6 Da fragment mass tolerance, a maximum of three missed cleavage sites, dynamic modification of 15.995 Da for methionines, static modifications of 113.084 Da for cysteines, and 304.207 Da for peptide N-termini and lysines. The Percolator algorithm was applied to the Sequest HT results. The peptide false discovery rate was set to 1% and the protein false discovery rate was set to around 5%. TMT reporter ions of the MS3 spectra were integrated with a 20 ppm tolerance and the reporter ion intensities were used for quantification.

The mass spectrometry proteomics data have been deposited to the ProteomeXchange Consortium (http://proteomecentral.proteomexchange.org) via the PRIDE partner repository[51] with the dataset identifier PXD040291 (DOI 10.6019/PXD040291) and PXD046677 (DOI 10.6019/PXD046677). Protein relative quantification was performed using an in-house developed R (v.4.2) script, available on GitHub (https://github.com/Novartis/px_tmt_daa). This analysis included multiple steps; (1) data filtering (exclusion of peptides mapping to multiple proteins, exclusion of PSM where the number of SPS mass matches were <60%, the precursor interference was >50% or the average reporter ion s/n <10, as well as exclusion of PSMs with missing reporter ion signals); (2) global data normalization by equalizing the total reporter ion intensities across all channels, (3) summation of reporter ion intensities per protein and channel, calculation of protein abundance log2 fold changes (L2FC) and testing for differential abundance using moderated t-statistics[52] where the resulting $p$ values reflect the probability of detecting a given L2FC across sample conditions by chance alone. Subsequently, the $p$ values were adjusted for multiple testing using the Benjamini–Hochberg method ($q$ values).

### Western blotting and co-immunoprecipitation

AK59 and QK50 compound treatments were always kept at 16 h incubation and 10 µM concentration unless otherwise indicated. Proteasomal inhibition (bortezomib Sigma-Aldrich, Cat#504314, diluted in 0.1% DMSO final, MG132 Selleckchem, Cat#S2619) or neddylation inhibition (MLN4924, Selleckchem, Cat#S7109) was initiated 1 h prior to compound treatment and maintained during the 16 h of AK59 incubation.

Cells were pelleted and washed with 1x DPBS (Thermo Fisher Scientific, Cat#14190094) before lysis. For western blot analysis, cell pellets were lysed with 5x extraction buffer (from co-immunoprecipitation kit, Thermo Fisher Scientific, Cat#14321D), diluted to 1x with 1x DPBS (Thermo Fisher Scientific, Cat#14190094) and supplemented with cOmplete protease inhibitor (Sigma-Aldrich, Cat#CO-RO) for 30 min on ice with pulse-vortexing every 5 min. To remove cell debris, lysate was spun at $15,000 \times g$ at 4 °C for 15 min. The supernatant containing proteins were collected in a fresh tube, and protein quantification was done using Pierce BCA assay (Thermo Fisher Scientific, Cat#23225) according to the manufacturer's protocol. About 20 µg of protein/well in 20 µl loaded for each sample and then prepared with NuPAGE™ LDS sample buffer (Thermo Fisher Scientific, Cat#NP0007) and NuPAGE sample reducing agent (Thermo Fisher Scientific, Cat#NP0004), boiled in 70 °C for 10 min, loaded on pre-cast either 10-well, 12-well, or 15-well NuPAGE™ 4 to 12% Bis-Tris 1.5 mm mini protein gel (Thermo Fisher Scientific, Cat#NP0335BOX, NP0322BOX, and NP0336BOX respectively). A protein ladder (Thermo Fisher Scientific, Cat#26619) was used to determine protein size. Samples run in 1x MES SDS running buffer (Thermo Fisher Scientific, Cat#NP0002) for 40–50 min at 200 V. Semi-dry transfer was performed using Trans-blot turbo transfer (Biorad, Cat#1704156) with ready-made PVDF membrane-containing transfer packs according to the manufacturer's protocol. Primary antibodies used in this study were all anti-human: STING (1:1000, CST, Cat#13647 or 1:500, Thermo Fisher Scientific, Cat#MA526030), UBIQUITIN (1:500, CST, Cat#3936), HERC4 (1:500, abcam, Cat#ab856732, batch number GR3184017-12), UBA6 (1:1000, CST, Cat#13386), UBA5 (1:500, abcam, Cat#ab177478), phospho-IRF3 (1:1000, abcam, Cat#76493), phospho-TBK1 (1:500, CST, Cat#5483), SMO (1:500, abcam, Cat#ab236465) and PGR (1:250, CST, Cat#8757). As loading control, ACTIN (1:500, Sigma, Cat#A5441), TUBULIN (1:500, CST, Cat#2146), and VINCULIN (1:500, CST, Cat#13901) were used, as described in figure legends. For secondary antibodies, HRP-conjugated anti-mouse (1:2500, CST, Cat#7076) and anti-rabbit (1:2500, CST, Cat#7074) antibodies were used. For detection, either Amersham ECL prime western blotting detection reagent (Cytiva Life Sciences, Cat#RPN2232) or SuperSignal™ West Femto reagent (Thermo Fisher Scientific, Cat#34094) were used according to the manufacturer's instructions. The ECL signal was visualized using Bio-rad ChemiDoc XRS+ and quantified using the Image Lab software (Bio-rad, Version 6.0.1). Each of the experiments was repeated in three independent biological replicates and one representative was shown in the figure.

HERC4 co-immunoprecipitations were performed using the Dynabeads™ Co-Immunoprecipitation Kit (Thermo Fisher Scientific, Cat#14321D) according to the manufacturer's protocol followed by a western blot (described above). About 50 µl of HERC4 antibody (Abcam, Cat# ab856732) conjugated with 10 mg Dynabeads™ (Thermo Fisher Scientific, Cat#14301), and for each pulldown sample, 1.5 mg of antibody conjugated Dynabeads™ were used with 50 µg of protein. The loading concentration of samples in western blots followed by co-immunoprecipitation was 50 µg protein/well in 20 µl. As a control, an equal amount of input or unbound fraction is run with pulldown samples. Each of the experiments was repeated in three independent biological replicates and one representative was shown in the figure.

Co-immunoprecipitations for ubiquitin pulldowns were performed using the Ubiqapture-Q kit (Enzo Life Sciences, Cat#BML-UW8995) according to the manufacturer's protocol. To detect captured ubiquitin levels, instead of the provided antibody from the kit, the UBIQUITIN antibody (1:500, CST, Cat#3936) was used. About 50 µg of protein was used to pulldown. As a control, equal amount of input or unbound fraction run with pulldown samples. Each of the experiment repeated in three independent biological replicates and one representative was shown in the figure.

## IRF pathway reporter assay

In order to measure IRF pathway activity, an IRF-Lucia luciferase reporter system containing Dual-THP1-Cas9 cells (Invivogen, Cat#thpd-nfis) were used. IRF pathway reporter assays were performed in 96-well plates (TPP, Cat#ZZ707902) where Dual-THP1-Cas9 cell line derivative cells were seeded at a cell density of 0.4 million cells/well and stimulated with 30 μM cGAMP (Biolog, Cat#C161) immediately. After 3 h stimulation, cells were treated with the indicated compounds, and the luciferase measurement was performed 16-h after compound incubation. To detect luciferase activity, QUANTI-Luc™ (Invivogen, Cat#rep-qlc4r1) was prepared according to the instructions on the datasheet. The reagent (50 μl) was added to a black with clear bottom 96-well plates (Greiner, Cat#655090) where the bottom of the plate was sealed (PerkinElmer, Cat#6005199). About 20 μl of the Dual-THP1 cells from each condition pipetted onto the luminescence reagent in a 96-well plate and luminescence was measured for 0.1 s with the EnVision® Multimode plate reader (Revvity, workstation version 1.14.3049.1193). Each of the experiment repeated in three independent biological replicates. Fold increase in luminescence was calculated by dividing each read to its matching control group (DMSO treated) and results were plotted using Graphpad Prism 9.

## HTRF assay

For the HTRF assay, wildtype, Ctrl, or HERC4 sgRNA transduced Dual-THP1 were seeded in 200,000 cells/well in V-bottom 96-wells (Corning, Cat#3894). Cells were stimulated with 30 μM cGAMP (Biolog, Cat#C161) right after seeding. After 3 h stimulation, cells were treated with indicated doses of AK59, QK50, or DMSO control. After 16 h compound treatment, cells were spin down at 1200 rpm for 3 min and supernatant was taken to proceed with HTRF assay. CXCL10 and IFN-beta cytokine measurements from cell supernatants were performed by HTRF CXCL10 kit (Cisbio, Cat#62HCX10PEG) and HTRF IFN-beta kit (Cisbio, Cat#61HIFNBPEG) according to manufacturer's instructions. Briefly, standards for each assay are prepared fresh according to the indicated concentrations. For each measurement, pipetted in triple technical replicates in 384-well white low-volume plates (Greiner, Cat#784075). In order to avoid split over of fluorescence, samples were loaded with one well gap in between. Each supernatant sample was pipetted in triplicate technical replicates. The premix of antibodies for each assay was prepared according to the instructions and added on top of standards as well as samples. About 665 and 620 nm measurements were done using an EnVision® Multimode plate reader (Revvity, workstation version 1.14.3049.1193). Delta ratio and %CV are calculated according to the formula provided by the manufacturer. Standard curves were plotted for each assay, and unknown sample measurements were interpolated from the standard cure using GraphPad Prism 9 software. Each assay was performed in three biological replicates, and results were plotted using GraphPad Prism 9 software.

## PBMCs isolation from whole blood

Anonymized whole blood was collected under informed consent from healthy volunteers through the Interregionale Blutspende (IRB) of the Swiss Red Cross (SRK) in Bern, Switzerland. 2x 9 ml of whole blood taken from four independent donors. Independent donors were used as biological replicates of the data. About 18 ml blood diluted with 18 ml 1x DPBS (Thermo Fisher Scientific, Cat#14190094) pipetted slowly in SepMate tubes (Stemcell Technologies, Cat#15460) which were already balanced with 15 ml Lymphoprep solution (Axis-Shield, Cat#1114547). Samples were centrifuged at 1200 × g for 15 min with slow brake, and then the top and middle layers were carefully separated. The platelets were removed with a centrifuge at 114×g for 10 min. red blood cells were removed by RBC lysis buffer (Thermo Fisher Scientific, Cat#00-4333-57). Before seeding for experiments,

cells were strained with a 70-μm cell strainer (BD, Cat#352350). For each compound condition, 1 million cells/well in 24-well plates (TPP, Cat#Z707791) were seeded. Samples were treated with either 10 μM of AK59, QK50, or DMSO for 16 h. For STING quantification on human PBMCs after AK59 treatment, cells were treated with 1:50 diluted Fc block (BD, Cat#301804) for 10 min at room temperature. After the block, cells were stained with CD14 antibody (Biolegend, Cat#325611) in 1:50 dilution for 30 min at room temperature. Wash the cells using FACS Wash Buffer containing 1x DPBS (Thermo Fisher Scientific, Cat#14190094) and proceed with the fixation step of the "STING quantification by flow cytometry" protocol.

For analysis, after cells gated for single cells by forward and side-scatter, CD14+ cells were gated for monocytic population in PBMCs and gated group of cells were then further analyzed for STING expression.

## STING quantification by flow cytometry

THP1-Cas9 cells were seeded at a density of $1 \times 10^6$ cells/ml in 24-well plates with the media containing the indicated compound. Compound incubation time varied between 5–16 h and indicated concentrations of bortezomib treatments were always 1 h prior to any additional compound treatment. Cells were always seeded together with the initial treatment. After the compound incubation, cells were collected and fixed using 2.5% paraformaldehyde (stock 32%, Electron Microscopy Sciences, Cat#15714-S) at 37 °C for 10 min. Cells were then washed using FACS Wash Buffer containing 1x DPBS (Thermo Fisher Scientific, Cat#14190094) + 0.5% FBS + 2 mM EDTA and permeabilized at room temperature for 20 min using 100 μl of Perm/Wash I (BD, Cat# 557885), diluted 1:10 with 1x DPBS (Thermo Fisher Scientific, Cat#14190094). Cell washing was performed again and then samples were stained with 150 μl/sample anti-STING Alexa488; 1:200 (Abcam, Cat# ab198950) diluted in Robosep Buffer, which contained PBS + 2.0% FBS + 1 mM EDTA (Stemcell; # 20104) at 4 °C for 1–2 h. After antibody incubation, cells were washed with Wash Buffer three times; then cell pellets were resuspended in FACS Wash Buffer, and flow cytometry acquisition on Fortessa was then performed using Diva software (BD, version 9.0.1). Each of the experiment repeated in three independent biological replicates. Analysis was performed using FlowJo software (version 10.6.1).

## Visualization and representation of STING crystal structure

STING structure published by ref. 53 was used, accessed from the PDB database (Entry: 7SII). STING structure was rendered in the PyMOL Molecular Graphics System (version 3.0 Schrödinger, LLC). SAVI mutation information was taken from previous reports[36].

## STING-GFP construct design and tracking with live cell imaging

Either full-length or cytosolic domains for STING (consisting of either the C-terminal 141–341aa or 155-341aa) were cloned into the pcDNA3.1(+) vector backbone with and without a C-terminal GFP tag. HEK293T cells were seeded on six-well plates (TPP, Cat#Z707759) with a cell density of 1 million cells/well. For live tracking of the protein expression, 1 day after seeding, cells were transfected with the GFP-tagged STING expression constructs using the Trans-IT transfection reagent (Mirus, Cat#MIR2304). After transfection, cells were placed into an Incucyte® (Sartorius, version 2022B rev2) live cell imager where 10x phase contrast and GFP images were taken. Expression of GFP-tagged constructs were observed over 48-h from the start of the transfection until treatment. A total of 16 images from each well were recorded for better coverage, in 2-h time intervals. GFP+ cell area normalized to total cell area per well. The GFP channel was set to identify GFP+ in the cut-offs of minimum 0 and maximum 1000 as a default setting where the background is subtracted.

After 48-h transfection, cells were treated with either DMSO or 10 μM AK59 for 16 h. During treatment the time interval decreased

30–45 min. Data normalized to the start of the treatment and plotted as the delta GFP (treatment-control). For the HERC4 knockout and Ctrl sgRNA comparison, all the wells were treated with the AK59, and the GFP signal difference between STING[155-341] and STING[141-341] was measured. Data normalized to the start of the treatment and plotted as the delta GFP (STING[141-341]- STING[155-341]). Each of the experiment repeated in three to four independent biological replicates.

For each STING-GFP live imaging experiment, complementary western blot analysis was performed on HEK293T cells transiently transfected with non-GFP-tagged STING constructs in order to decrease background in blots. Each of the experiment was repeated in three independent biological replicates and one representative was shown in the figure.

## CRISPR genome-wide screening

For genome-wide CRISPR knockout screening, THP1 cells constitutively expressing Cas9 were generated by lentiviral delivery of the Cas9 protein gene in pNGx-LV-c004 and selected with 5 µg/ml blasticidin S HCl (Thermo Fisher Scientific, Cat#A1113902) as previously described[54]. For screening, we used a sgRNA library targeting 18,360 protein-coding genes with five sgRNA/gene[55]. The sgRNA library was packaged into lentiviral particles using HEK293T cells as previously described[55,56]. Briefly, $2.1 \times 10^7$ HEK293T cells were seeded into Cell-STACK (Corning, Cat#3391) cell culture chambers and transfected with the sgRNA library plasmid mix together with ready-to-use lentiviral packaging plasmid mix (Cellecta, Cat#CPCP-K2A) 24 h after seeding using the Trans-IT transfection reagent (Mirus, Cat#MIR2304). Viral particles were harvested 72 h post-transfection and quantified using the Lenti-X qPCR kit (Clonetech, Cat#631235).

THP1-Cas9 cells were expanded for library transduction. On day zero, the cells were seeded and transfected to achieve a coverage of the library of at least 1000 cells/sgRNA with a multiplicity of infection (MOI) of 0.5. 5 µg/ml polybrene (Millipore, Cat#TR-1003-G) was used in transfection of THP1-Cas9 cells. Transduced cells were selected with 4 µg/ml puromycin for 3 days and on the 4th day, cells were analyzed for RFP expression using the FACS Aria (BD) for determining transduction efficiency. After collecting the day-4 samples, the rest of the library-transduced cells were seeded for the screen with two biological replicates per condition. At day 10, cells were treated with either DMSO or 10 µM AK59 and incubated for 16 h. After compound incubation, cells were harvested, fixed, and stained for STING expression (described in detail below). Then the fraction of cells with the 25% highest and 25% lowest levels of staining for STING, in each treatment group, the sorted samples were processed for genomic DNA isolation using the QIAamp DNA blood maxi kit (Qiagen, Cat#51192) according to the manufacturer's protocol. Genomic DNA was quantified using the Quant-iT PicoGreen assay (Thermo Fisher Scientific, Cat#P7589) according to the manufacturer's recommendations and proceeded with Illumina sequencing.

## Illumina sequencing of the library

The integrated sgRNA sequences were PCR amplified using primers specific to the integrated lentiviral vector sequence and sequenced using the Illumina sequencing technology. Illumina library construction was performed as previously described[55]. Briefly, a total of 96 µg of DNA per sample was split into 24 PCR reactions, each with a volume of 100 µl, containing a final concentration of 0.5 µM of each of the following primers (Integrated DNA Technologies, 5644 5′-AATGATAC GGCGACCACCGAGATCTACACTCGATTTCTTGGCTTTATATATCTTG TGGAAAGGA-3′ and INDEX 5′-CAAGCAGAAGACGGCATACGAGATXXX XXXXXXXGTGACTGGAGTTCAGACGTGTGCTCTTCCGATC-3′, where the Xs denote a ten base PCR-sample specific barcode used for data demultiplexing following sequencing), 0.5 mM dNTPs (Clontech, Cat#4030), 1x Titanium Taq DNA polymerase, and buffer (Clontech, Cat#639242). PCR cycling conditions were as follows: $1 \times 98\,°C$ for

5 min; $28 \times 95\,°C$ for 15 s, 65 °C for 15 s, 72 °C for 30 s; $1 \times 72\,°C$ for 5 min. PCR samples were purified using 1.8x SPRI AMPure XL beads (Beckman Coulter, Cat#A63882) according to the manufacturer's recommended protocol and the qPCR quantified using primers specific to the Illumina sequences according to the sequencing library qPCR quantification guide (Illumina, Cat#SY-930-1010). Amplified libraries were then pooled and sequenced with HiSeq 2500 instrument (Illumina) with 1x 30b reads, using a custom read 1 sequencing primer: 5645 (5′-TCGATTTCTTGGCTTTATATATCTTGTGGAAAGGACGAAA-CACCG-3′), and a 1x 11b index read, using the standard Illumina indexing primer (5′- GATCGGAAGAGCACACGTCTGAACTCCAGTCAC-3′), according to the manufacturer's recommendations.

## Analysis of the CRISPR screen

Sequencing analysis was performed as previously described[55]. In short, raw sequencing reads were converted to FASTQ format using bcl2fastq2 (version 2.17.1.14, retrieved from http://support.illumina.com/downloads/bcl2fastq-conversion-software-v217.html), trimmed to the guide sequence with the fastx-toolkit (version 0.0.13, retrieved from http://hannonlab.cshl.edu/fastx_toolkit/index.html) and aligned to the sgRNA sequences in the library using bowtie[57] with no mismatches allowed. Differential presentation of sgRNAs was calculated using DESeq2[58] and gene-level results were obtained using the redundant siRNA activity (RSA) algorithm[59]. In RSA, the rank distribution of individual sgRNAs is examined to calculate a hypergeometric enrichment score for the concerted action of each gene's set of guides. This results in a gene-level p-value for significance. Furthermore, the lower (Q1) or upper (Q3) quartile of the sgRNA's fold changes to represent effect size at the gene level.

Due to the pooled analysis approach in the RSA algorithm, the range of RSA and Q values in each comparison is in a similar magnitude which enabled further comparisons. In order to rank the hits, RSA and Q values from two different comparisons (in this case, DMSO high vs. low STING and AK59 high vs. low STING) were taken as point coordinates. The Euclidean distance (vectoral) between two coordinates (*a.k.a* RSA and Q values of each comparison) were then calculated as the magnitude of the vector and the direction of the vector was described by the increase/decrease of RSA and Q values between two points (Bioconductor 4.0.2, Supplementary Data 4). Then, hits were ranked according to the magnitude and the direction of their vector.

## STRING protein association analysis

Interaction between STING, HERC4, UBA5, and UBA6 was shown using STRING db. (Version 11.5 with the minimum required interaction score of "medium confidence (0.4)" or version 12 with the minimum required interaction score of "low confidence (0.15)")[38].

## Individual CRISPR knockouts and TIDE analysis

THP1-Cas9, Dual-THP1-Cas9[40], and HEK293-JumpIN-Cas9 cells were generated by lentiviral delivery of the Cas9 protein gene in pNGx-LV-c004[54] and selected with 5 µg/ml blasticidin S HCl (Thermo Fisher Scientific, Cat#A1113902). Individual knockouts were generated by the lentiviral delivery of sgRNAs in the pNGx-LV-g003 backbone. SgRNA transduced cells were selected with 3.5 µg/ml puromycin (Thermo Fisher Scientific, Cat#A1113802). sgRNA sequences targeting the gene of interests are listed below. Knockout efficiency was checked after the third passage by both Tracking of Indels by Decomposition (TIDE) analysis[39] and Western blot.

To check the rate of Indel formation at the CRISPR cut-site, TIDE analysis[39] was used as previously described[60]. Briefly, genomic DNA was extracted from approximately one million cells per condition using a DNA extraction kit (Qiagen, Cat#69504) according to the manufacturer's recommendations. DNA concentration was measured with the Nanodrop (Thermo Fisher Scientific), and PCR reactions were performed with 2x Phusion polymerase master mix (Thermo Fisher

Scientific, Cat#F548S), 5% DMSO, 100 ng of DNA and a final concentration of 1 μM forward and reverse primer. PCR primers for each gene are given in Supplementary Table 1. The amplification run was as follows: initial denaturation at 98 °C 30 s, 30 cycles of denaturation at 98 °C for 5 s, annealing at 61 °C for 10 s, extension at 72 °C for 15 s and final extension at 72 °C for 2 min. PCR samples were cleaned up using PCR and Gel extraction kit (Qiagen, Cat#28704) according to the manufacturer's protocol and Sanger sequenced. Frameshift, in frame and wildtype calculated from the TIDE analysis results and undefined sequencing reads were excluded.

### NanoBiT® complementation assay
Full-length human HERC4 (CCDS41533) was cloned into the pFN35K SmBiT TK-neo Flexi® Vector, encoding an N-terminal small bit (SmBiT) tagged construct (Promega) with the native HSV-TK promoter swapped out for a CMV promoter. C1025A mutation was introduced with site-directed mutagenesis. The cytosolic domain of human STING (either 141–341, 155–341, or 141–341_K150R) was cloned into the pFC34K LgBiT TK-neo Flexi® Vector, encoding a C-terminal large bit (LgBiT) tagged construct (Promega) which is transcribed under the control of the HSV-TK promoter.

Next, $4 \times 10^4$ HEK293T cells were seeded into 96-well plates (TPP, Cat#ZZ707902). On the same day, cells were transiently transfected with both constructs using a Trans-IT transfection reagent (Mirus, Cat#MIR2304) according to the manufacturer's recommended protocol. Then 48 h after transfection, cells were treated with the indicated doses and incubation times of either DMSO, AK59, or QK50. For the dose-response, all samples were measured at 16-h post-compound treatment. For the QK50-AK59 competition, QK50 (or DMSO control) treatment started 3 h prior to 16-h AK59 treatment with the indicated doses. For the cGAMP competition with AK59, cGAMP stimulation was initiated by the addition of 30 μM of cGAMP three hours prior to the 16-h compound treatment. To detect the luciferase activity, the Nano-Glo® luciferase assay system (Promega, Cat#N1110) was used according to the manufacturer's protocol and luminescence was measured for 0.1 s with EnVision® Multimode plate reader (Revvity, workstation version 1.14.3049.1193). Each of the experiment repeated in three to four independent biological replicates. Fold increase in luminescence was calculated by dividing each read to its matching control group (DMSO treated) and results were plotted using Graphpad Prism 9.

### Statistics
Statistical analysis of the indicated data was performed using Graph-Pad Prism 9. Data points were represented as the mean ± standard deviation (SD). Appropriate statistical tests for each data were indicated in the figure legends. Significance indicated as ns $p > 0.05$, *$p < 0.05$, **$p < 0.01$, ***$p < 0.001$, ****$p < 0.0001$. Additionally, exact $p$ values were indicated wherever it is possible.

### Reporting summary
Further information on research design is available in the Nature Portfolio Reporting Summary linked to this article.

## Data availability
The proteomics data generated in this study have been deposited in the PRIDE database under accession codes PXD040291 and PXD046677. Additionally, full list of proteomics results are provided in Supplementary Data 1, 3. Crystal structure of STING was previously published by ref. 53 (PDB ID: 7SII). The raw data and the analysis of CRISPR screen data generated in this study have been provided in Supplementary Data 2. Vectoral ranking code is provided with this paper in Supplementary Data 4. Source data are provided with this paper.

## Code availability
Protein relative quantification was performed using an in-house developed R (v.4.2) script, available on GitHub [https://github.com/Novartis/px_tmt_daa] (https://doi.org/10.5281/zenodo.10962720).

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

## Acknowledgements

We thank the BR Discovery Postdoctoral program for support of M.M. Work in this manuscript has been funded by the Novartis Research Foundation.

## Author contributions

M.M., I.S., B.G., F.F., A.H., C.N.P., and D.G. designed the research. M.M., I.S., A.I.M., B.G., F.F., D.Be., O.S., S.K., S.R., D.Bo., D.J., T.T., R.M., S.A., and

A.R. performed research. M.M., N.P., E.A., and A.H. analyzed data. M.M., F.N., A.H., J.R.H., C.N.P., and D.G. wrote and edited the manuscript.

## Competing interests

M.M., I.S., B.G., F.F., D.B., O.S., N.P., E.A., S.K., S.R., D.J., T.T., R.M., S.A., A.R., F.N., A.H., C.N.P., and D.G. are current employees and shareholders of Novartis Pharma. A.I.M., D.Bo., and J.R.H. are former employees of Novartis Pharma. D.Bo. is a current employee of Monte Rosa Therapeutics, Basel, Switzerland., J.R.H. is a current employee of Vector Biology, Cambridge, MA, USA.
