## [Peer Review File · Nature Communications]

Small molecule induced STING degradation facilitated by the HECT ligase HERC4REVIEWER COMMENTS

Reviewer #1 (Remarks to the Author):

Mutlu et al., describe development of a small molecule AK59 which acts as a HERC4 E3 ubiquitin ligase dependent degrader of STING. STING is a major element in innate immune system as part of cGAS-STING pathway and its inhibition of this pathway for autoimmune diseases have attracted considerable interest. The authors identified AK59 in a phenotypic screen for type 1 interferon inhibition and further validated it to downregulate STING. CRISPR screen identified HERC4 as a E3 ubiquitin ligase which could be responsible.

Overall study is well designed and identification of novel molecular glue degraders (while the mode of action remains to be fully elucidated) is of great interest to the TPD field.

There are few questions and clarifications that are need addressing prior to publication.

1. Could the authors comment on general toxicity of AK59 or QK50 in THP1 cells at 24 or 3 day timepoint - was that assessed?

2. Paragraph in line 138-158. Mutation of lysine 150 to arginine. The authors identified the STING 141-155 region as critical for degradation, mutate lysine 150 to arginine, and then reason that that mutation is essential for ubiquitination of STING induced by AK59. Another, explanation would be that K150 is part of a 'degron' motif, or perhaps interacts with the AK59 directly, or is at the interface between HERC4 and STING. I would question the reasoning here - mutation could also reduce the binary binding affinity, or prevent ternary complex formation which would also affect ubiquitination.

3. Line 171/296 - I offer my skepticism on the identification of a lysine residue (from ubiquitination point of view) - please see comment above. That would need to be further validated - perhaps ternary complex, or compound binding is inhibited instead? NanoBit assay STING S150K:HERC4 would be able to answer this question.

4. Line 268 and Extended data 6a. It appears that STING 155-341 pulls down with AK59, but longer construct which includes the helix STING 141-341 does not. In figure 2d - degradation data - it appears that STING 155-341 is not degraded by AK59, but 141-341 is. This appears to be contradictory. Why would STING-141-341 not interact with HERC4 upon compound treatment, but be potently degraded?

5. Did the authors consider testing the negative compound QK50 in a competition experiment with the AK59 in a nanobit assay? If successful, cells pre-treated with constant amount of AK59, with a dose response of QK50 could show competition, indicating that QK50 maintains binary binding to either STING or HERC4, but the quinoline in QK50 affects the formation of ternary complex specifically.

Minor points:

Line 66. Please provide references for the expression level association with these cancers - is it over-expression or loss of ligase?

Fig 2c and d. or 5a The representation is rather confusing. It may be easier for the reader to have a downward trend to look at - synonymous with degradation. Perhaps a normalization to DMSO would allow for this?

Fig 3d. Please consider labeling other top factors identified in the screen.

Reviewer #2 (Remarks to the Author):

STING is a protein associated with many inflammatory diseases. While the understanding of STING is far from complete, removal of STING by genetic approaches or inhibition of STING by small molecule inhibitors can ameliorate disease.

This study utilized several approaches and strategies to describe the identification of AK59-51B (abbreviated AK59) as a small molecule which induces STING degradation. Broadly speaking, the two main cell types utilized in this study are THP1 and HEK293. AK59 was first identified as a small molecule that type I interferon response upon exogenous DNA stimulation (this data was not shown). Different forms of STING were expressed in THP1 and HEK293 cells to assess the impact that AK59 has on the levels of STING. Furthermore, a CRISPR screen was employed in THP1 cells, enabling the identification of HERC4 as a mediator of STING degradation upon AK59 treatment. Several genetic knockout cell lines were used to complement the finding of the CRISPR screen. Interestingly, HERC4 and STING associate based on a luciferase complementation assay and immunoprecipitation experiments. The findings are interesting and relevant given the broad relevance of the STING pathway and targeted protein degradation.

The findings of the study are important to enable the discovery of new mechanisms to modulate STING levels for treatment of STING-associated diseases. The authors have used independent methods to generate data for their study. However, I highly recommend that fewer but well-designed experiments should be the focus of the authors to improve the quality of the manuscript, rather than executing too many different experiments with flaws in the quality of the data. In addition, a major concern is that most of the manuscript contains inaccurate interpretation of data either due to uncontrolled experimental design or inadequate information. It is still difficult to conclude whether AK59 acts as a molecular glue that generates a new interface between HERC4 and STING to then degrade STING. I recommend that the authors reconsider removing experiments and adding new experimental evidence. I believe the manuscript should be rejected from acceptance at Nature Communications unless the gaps and flaws of the experiments are considered.

The major comments for the manuscript are described here. Experiments and comments for specific sections are grouped by sub-numbering (1-1, 1-2, 1-3, 1-4).

Point 1: How well does AK59 degrade STING?

1-1: A time course of AK59 inducing STING degradation is needed along with concentration doses for both Figure 1 and/or extended Figure 1. This is important for 2 main reasons; (a) this is important for understanding the behavior of the compound and, (b) it is unclear why the current concentration and time is chosen for AK59. Treating for 16 h with 10 μ M of AK59 seems too long to induce STING degradation. It is unclear if this prolonged 16 h incubation will result in off-target cellular effects such as induction of cellular stresses or rewiring or cellular signaling, therefore causing STING degradation. This data will fit into figure 1 or extended figure 1.

Point 2: Does AK59 induce ubiquitination of STING?

2-1: The ubiquitin-based pulldowns can be improved to make the claim that STING is ubiquitinated in the presence of AK59. In figure 1g, 2b, 5c, it is not stated how long AK59 is present for. Please state this clearly as it influences interpretation. In figure 1g, lanes 3 and 4 do not appear different from lanes 1 and 2. I would expect 50 nM bortezomib to protect STING against AK59-induced degradation of STING, but this does not look to be the case compared to lanes 1 and 2 (see input blot). Furthermore, I would expect that ubiquitin pulldown to show more "ubiquitinated STING" (lane 4) upon AK59 treatment with bortezomib compared to AK59 without bortezomib (lane 2) (see IP: ubiquitin). Please provide an explanation for this. As AK59 degrades STING, less ubiquitinated STING should be available in the ubiquitin pulldown (lane 2) compared to if AK59 and bortezomib were both present (lane 4).

2-2: Following on from point 2-1, given the authors mostly utilize cell-based assays, it would be important to provide more evidence that AK59 drives degradation of STING through a ubiquitin signal of

relevance (K48-linked polyubiquitin chains for example). This could be done by recombinant protein experiments or other types of cell-based experiments. One way to do this is to transfect the classical HA-ubiquitin constructs (WT, K48R, K63R) into either HEK293 or THP1 cells followed by 10 μ M AK59 treatment (or vehicle) in the presence or absence of 50 nM bortezomib. A pulldown with anti-HA followed by detecting for STING will provide evidence if AK59 drives degradation by STING through a K48-linked ubiquitin signal. This provides a mechanism of action for the claimed molecular glue. The authors can repeat the experiment in HERC4 KO cells if needed.

Point 3: Quality of data for STING constructs in Figure 2 requires major revisions.

3-1: The experiments in Figure 2 for the section “Lysine 150 of STING is essential for AK59-mediated STING degradation” is made up of a mix of experiments that do not seem to be controlled and contain inconsistent results from the rest of the manuscript. I would suggest removing this entire figure and entire section as it does not add value. If the authors wish to keep this section, I recommend changing the interpretation or redoing the experiments according to the other points below (3-2 to 3-11).

3-2: In Figure 2a, please also include experiments for wild-type full-length STING tagged with GFP as this best represents endogenous STING for consistency with Figure 1. Do the truncated versions / GFP-tagged versions get degraded faster than endogenous STING? This seems to be the case because maximal degradation occurs at 2 – 4 h for construct 1 (STING 141-341). Important to also relabel the legend to ensure authors note the construct number (construct #1, construct #4) AND whether the protein is GFP tagged or not.

3-3: Why is the fraction of GFP+ cells ranging from 0.9 to 1.3 in the scale only? This seems to suggest the degradation is very partial in these assays. For example, 1.0 to 0.9 suggests a 10% reduction of GFP positive cells. Can the authors really claim that this STING 141-341-GFP is degraded by AK59?

3-4: Figure 2b requires input blot for STING and control with bortezomib. What is the basal level of STING 155-341, STING 141-341, STING 141-341 (K150R)? At this point, if the expression levels from the HEK293 transfection varies from construct to construct, authors cannot claim anything about relative ubiquitination levels or status between constructs, please revisit the statement “ubiquitination levels remained constant in either STING 155-341 or STING 141-341 K150R” (see line 155). Importantly no mention of treatment time is indicated. Is this a 16 h experiment as well? How is it possible to immunoprecipitate (pull down) ubiquitinated form of STING if AK59 is triggering substantial degradation of STING constructs? To confidently conclude that ubiquitinated form of STING is captured, authors need to redo the experiment in figure 2b with 50 nM bortezomib and a vehicle control to observe differences.

3-5: Figure 2c data should be plotted as in Figure 2a. Replot as fraction of GFP+ cells (normalized to treatment start). In the current fold change graph, it is difficult to compare the findings to figure 2a. For

example, in figure 2c, it is interpreted that STING 141-341 (construct #1) maximal degradation occurs at 12 – 16 h, whereas in figure 2a it isn't the case.

3-6: Figure 2a and 2c should be complemented with cell confluence data in the IncuCyte. This is to ensure cell numbers are not changing across time during the experiment. Please provide this data. In the current form, figure 2a and 2c can easily be affected by variability of IncuCyte plating density.

3-7: Figure 2a and 2c should be complemented with GFP intensity measurements. Rather than just counting GFP positivity (yes GFP present, or no GFP is absent), it is important to report GFP mean intensity from the IncuCyte as it resolves high GFP to low GFP signal. This gives another important layer of information on how fast acting AK59 is in terms degrading STING-GFP constructs.

3-8: In figure 2, it is not clear how many times the experiments were done. It seems the experiments in the entire manuscript were only done once. Please state clearly with the following statements: (a) "Western blot is representative of 3 independent experiments", (b) "The IncuCyte experiments were carried out in 3 replicates and repeated in 3 independent experiments, the graph shows mean \pm standard deviation from the replicates of 1 experiment". This comment is highlighted in in minor comments as well as a general note for methods reporting in figures.

3-9: Are the constructs express to the same extent or variably? If there is too much STING expressed, can AK59 degrade it? In figure 2, it is essential to provide an assessment of STING protein expression by Western blotting or mean GFP fluorescence intensity by FACS. A quantification of relative protein levels by Western blot or FACS will give an idea whether the STING constructs express similarly. This is important to report variability in experiments and how much impact does STING protein levels have on efficacy of AK59 to degrade STING. Please address this technical consideration point by experiments.

3-10: STING variants are claimed to be "partially degraded" in figure 2d and 2c. In figure 2d, how much degradation is occurring? Provide quantification of the Western blot in figure 2d this reduction in STING V155M and N154S by AK59 appears to be variation in loading. Also, in figure 2c, the fold change from 0 to 0.2 for STING V155M and N154S could be due to variability in cell density or cell growth. Please plot additional data as in points 3-6 and 3-7 above to provide cell confluence and GFP intensity data as well.

3-11: Overall, figure 2 requires substantial rework and additional experiments to support the title "lysine 150 of STING is essential for AK59-mediated STING degradation". The most fundamental experiment is to express STING full length (WT) and STING full-length (K150R) in HEK293T followed by 10 μ M AK59 or vehicle treatment in the presence of bortezomib or a vehicle control. This experiment alone could support the title well.

Point 4: The role of HERC4 in AK59-induced degradation of STING

4-1: The authors carried out a successful genetic based screen to identify regulators of AK59-induced degradation of STING, testing ubiquitin-related proteins from their top hits. HERC4 knockout provided only partial protection against AK59-induced degradation of STING in figure 4, why might this occur? Please provide an explanation in the discussion or results. Could it be possible that the other top hits (FASTKD5, PMAIP1, UQCRFS1) have a greater role in AK59-induced degradation of STING?

4-2: Is the E3 ligase activity of HERC4 needed for AK59-induced degradation of STING? This will be important to address. Otherwise, please remove the phrasing "E3 ligase HERC4" throughout the text (line 237, line 246 and anywhere else in the manuscript where needed). It's not clear at this point if HERC4 E3 ligase function is directly linked to degradation of STING by AK59. It would be informative to test if HERC4 WT compared to catalytic dead mutant overexpression will be favoring AK59-induced degradation of STING.

4-3: Figure 5a contains multiple flaws including insufficient data, contradictory data, unclear data. Firstly, figure 5a needs to have additional plots (confluence graph, GFP intensity) as it faces the same issues with earlier IncuCyte experiments. Next in this experiment, it is not clear which STING-GFP construct was used even though authors refer to extended figure 2. Please specify in materials and methods section, figure legend and results section. This is major information that needs reporting. This will affect the interpretation of data and will determine whether further experiments are needed. The other flaw to address is whether this is from one experiment or merged data from different experiments. The current data shows that if the standard deviation bars were plotted at both ends of the mean, both data for the black (control sgRNA) and red lines (HERC4 sgRNA) would overlap. This will then mean there's no change between ctrl sgRNA and HERC4 sgRNA.

4-4: Due to the insufficient data that HERC4 is mediating AK59-induced degradation of STING, authors need to generate more data. For example, this could involve HERC4 knockout in THP1 and HEK293T cells, followed by AK59 treatment and assessment of endogenous STING by Western blotting. A control such as 50 nM bortezomib is needed. It is important to also repeat such experiment with the STING constructs, specifically construct #1 (STING 141-341-GFP), construct #5 (STING 141-341). Can HERC4 knockout rescue AK59-induced degradation of STING and how does this compare to bortezomib treatment to block degradation?

4-5: It is unclear why data in figure 5b and figure 5c currently sit in the main figures as they do not add further understanding into how AK59 works as a potential molecular glue. It is however important to improve figure 5b as the quality of the data is questionable. Firstly, is reproducibility, in figure 5b, this is the first time that STING is completely absent with AK59 treatment, is this real reproducible result? I would suggest moving figure 5b to extended data and repeating the data to provide more repeats,

running multiple experiments on a single experiment. This is important to make the claim AK59 degrades STING partially or completely. Second is the finding that HERC4 disappears with AK59 treatment. It is important for this figure because the observation that HERC4 is degraded by AK59 is novel. One major issue is that the HERC4 band is noticeably below 115 kDa whereas in all other figures HERC4 is at least 115 kDa which corresponds to approximately the right size.

4-6: Figure 4c has the same issues as earlier immunoprecipitation experiments in figures 1g and 2b. There are multiple concerns and questions for this experiment that require further explanation or experiments. (1) An input blot for STING and HERC4 is essential here to demonstrate that AK59 is functioning as previous figures. (2) Also, the number of independent experiments must be reported. (3) How long is this experiment with the AK59 treatment? (4) Given AK59 triggers STING ubiquitination degradation, how is it possible to still immunoprecipitate (pull down) substantial amount of ubiquitinated STING (lane 3 and lane 4, STING blot)? (5) The experiment needs to be repeated as there is obvious differences in loading control (tubulin) which could lead to inaccurate interpretation of the immunoprecipitation. For example, authors claim that less of Smoothened was co-immunoprecipitated (lane 4) but this could be less starting material, evident by lower tubulin loading control.

4-7: Smoothened remains unchanged despite HERC4 loss with AK59. Smoothened is supposedly degraded by HERC4. In figure 5b and 5c, Smoothened loss is not observed with AK59 treatment. Why might this be the case? In figure 5 and extended data figure 6, the authors are trying to demonstrate that AK59 induces HERC4-STING complex formation and that AK59 interferes with HERC4-Smoothened complex formation. Thus far, the data have not been convincing. This is confounded by the fact that HERC4 itself is lost with AK59 treatment. One way to demonstrate this is to generate SmBiT-HERC4 and Smoothened-LgBiT constructs and express them in HEK293T cells. If AK59 generates a new interface to favor STING-HERC4 interaction, it would be expected that AK59 would reduce Smoothened-LgBiT/SmBiT-HERC4 interaction, which can be measured by luminescence. More experiments are needed to support the model in figure 5f. The model in figure 5f "naïve state" shows that HERC4 degrades Smoothened even though this is not the case in the hands of the authors.

4-8: Do STING truncated constructs interact with HERC4 upon 10 μ M AK59 treatment? Authors state that the STING constructs interact with HERC4 (lines 268 to 272) with AK59 treatment, but this is not shown convincingly. Extended data figure 6 should be repeated with proper controls, including DMSO controls. Authors also cannot make the claim in lines 271 and 272, that the connector helix and lysine 150 are essential for ubiquitination but not for formation of ternary complex in presence of AK59. To make such claim would require direct ternary complex formation experiments or mutagenesis experiments that abrogate HERC4-STING interaction.

4-9: Direct ternary complex formation between HERC4 and STING remains questionable. The authors have used a complementation luciferase assay, a conventional method to show protein-protein interaction. The authors claimed the connector helix of STING is required for degradation by AK59. Intriguingly, STING 155-341 (lacking connector helix) still interacts with HERC4 (extended data figure 6a). What is the minimal region required for STING-HERC4 interaction with AK59 treatment? One way to address this is by using the SmBiT-LgBiT assay but introducing mutations/truncations along the STING construct 141-341 in the 4 key helical regions (157-189, 263-274, 281-301, 325-338). Given cGAMP co-treatment does not interfere with STING-LgBiT/SmBiT-HERC4, the ligand binding pocket could be dismissed.

4-10: Mechanism of action and specificity of AK59 in cells and on HERC4 requires further clarity. While it is tempting to speculate AK59 is a molecular glue making STING a neo-substrate for HERC4, AK59 itself also decreases the levels of other proteins (figure 1d proteomics data). Moreover, HERC4 itself disappears with AK59 treatment. In the schematic proposed (figure 5f), AK59 acts on an interface within HERC4. It also acts between HERC4 and STING (figure 5f). This will be important to confirm and test given the proposed mechanism is that AK59 is a molecular glue. The specificity of AK59 is questionable. It could also be that AK59 creates a new interface between HERC4 and many other proteins (multiple neosubstrates), not just STING, which would argue against the specificity as suggested in the title of the manuscript "A novel HERC4-dependent glue degrader targeting STING". A proposed experiment is to revisit some of the proteins from proteomics in figure 1d and assess by Western blot whether some of the decreased proteins are also degraded by AK59. Intriguingly HERC4 is in the proteomics data of figure 1d with \log_2 fc of -0.72 and q-value of $1.63E-08$ but this was not reported. Alternatively, a structural / biochemical approach to show ligand binding of AK59 onto either HERC4, STING or both will be needed. Otherwise, the title of the manuscript should be rephrased to "A degrader targeting both STING and HERC4".

Point 5: Experiments needing clarity

5-1: Figure 1g, 2b, 5c have the ubiquitin band in the immunoprecipitation blots being inconsistent. Is the band size correct? Is the ubiquitin band below 30 kDa, 25 kDa or 15 kDa?

5-2: Figure 5d shows that AK59 does not degrade STING (input blot, lane 2 compared to lane 3). Why is this happening? HERC4 in the input blot also needs to be shown. It is also unknown how many times this experiment has been repeated.

The minor comments with data and/or data presentation are described below.

Point 1: Figure legend and figure labelling

1-1: In all figure legends, it is not clear how many times the experiments were done. Now, it seems the experiments in the entire manuscript were only done once. Please state clearly with the following statements: (a) "Western blot is representative of 3 independent experiments", (b) "The luciferase / IncuCyte / FACS experiments were carried out in 3 replicates and repeated in 3 independent experiments, the graph shows mean \pm standard deviation from the replicates".

1-2: In extended data figure 4, labelling for the legends has errors. I believe the labels need to correct to include DMSO for the last 4 conditions. All the treatments are currently AK59 and duplicated for some it seems.

Reviewer #3 (Remarks to the Author):

Key results:

The authors report the first molecular glue degrader that engages the HERC4 ligase. They also report the first molecular glue targeting STING, a high-value therapeutic target. The authors used a combination of proteomics, genomics and cellular assays to identify and validate the mechanism of action of their compound.

Validity:

Additional validation of the functional effect of compound-induced STING degradation is important. The manuscript includes data in Extended Data Figure 5 that shows a decrease in the activation of the IRF pathway in response to compound treatment; however, the data included also shows that there is a response in the HERC4 KO cells as well. The authors don't provide an explanation as to why there is a response in the HERC4 KO cells. Other publications reporting STING PROTACS, such as Liu, et al. J. Med. Chem 2022, demonstrate that their compound reduces levels of IFN- β , CXCL10 and IL-6 in cGAMP stimulated THP-1 cells. They also demonstrate that compound treatment decreases relative levels of phosphorylated TBK1. The authors should perform additional experiments that show similar effects on cytokine release and downstream target modulation. Additionally, those effects should be unique to wild-type THP-1 cells relative to the HERC4 cells or they should try to understand why there is a response in both wild type and HERC4 KO cells.

Originality and significance:

The field of targeted protein degradation is very active and there are many scientific research groups working to identify additional E3 ligases that can be therapeutically leveraged. The authors report the first targeted molecular glue degrader that engages the HERC4 E3 ligase. The validation of this E3 ligase opens up new avenues for researchers in the field and is of significant interest. STING is a highly validated therapeutic target that has been demonstrated to be involved in multiple inflammatory and autoimmune disorders. While there is research reported in the literature identifying inhibitors of the enzymatic activity of STING, there are only a few reports of alternative mechanisms of preventing STING activity in the cell. There is a 2022 publication reporting a PROTAC molecule that degrades STING via the CRBN E3 ligase; however, this manuscript reports the first molecular glue targeting STING. These findings are of significant interest to the field because of the novel E3 ligase targeted (HERC4), the more desirable physico-chemical properties of the molecule discovered (AK59-51TB) relative to the PROTAC that targets CRBN and the ability of AK59 to induce degradation of the SAVI mutants of STING.

Data & methodology:

-Discovery of AK59

The authors should include a detailed description of the screening methodology that was used to identify AK59. Currently there is a single sentence on lines 92-94 that gives no detail and no data is included. It would increase the significance and impact to the field if the authors included a description of the screening campaign with the screening assay used, the number of compounds screened, what the readout of the assay was, hit calling criteria, hit rate, hit confirmation method used and hit confirmation rate. The authors should also include a figure in the extended data section that illustrates the screening funnel and a scatterplot of the primary assay data showing the signal of all compounds tested, the threshold of hit calling and indication of where the hit related to AK59 is in that scatter plot.

-Role of Lys150 in STING degradation

The western blot data shown in Fig. 2b indicates that there is still some level of ubiquitination of the K150R mutant of STING, although at a reduced level relative to WT. The authors should provide a hypothesis as to why the K150R mutant is ubiquitinated.

Role of HERC4, UBA5 and UBA6 in compound induced STING degradation.

The authors performed a genome wide CRISPR screen to identify genes that were essential for AK59 induced STING degradation. They developed a FACS-based assay for monitoring the level of STING in cells. The resulting data convincingly show the essential role of HERC4 and UBA5 in controlling STING levels. The CRISPR data is less convincing for UBA6. The authors report that they were able to identify interactions between HERC4, UBA5 and UBA6 reported in the STRING database and cite that supportive evidence of the role they play. My search of the database did not find a reported interaction between any of the three proteins. My search included alternative names for HERC4 (KIAA1593), UBA5 (THIFP1, UFM-activating enzyme, UBE1DC1) and UBA6 (MOP-4, E1-E2, UBE1L2). The authors should include in the methods section a description of the search parameters used in the STRING database that will demonstrate those interactions. I was able to confirm that there was no interaction between STING and either HERC4, UBA5 or UBA6 reported in the STRING database.

The single gene KO studies provide convincing evidence that HERC4, UBA5 and UBA6 have a role in the level of STING in AK59 treated cells. The western blot data in Fig. 4(h) for sgRNA1 shows the band for the protein loading control, Vinculin, is significantly weaker than for the control and sgRNA2 samples. That experiment should be repeated so that the levels of Vinculin are the same for all three samples.

-AK59 is a MGD that induces an interaction between STING and HERC4

As mentioned previously, additional validation of the functional effect of compound-induced STING degradation is needed. Data reported in Extended Data Figure 5 is ambiguous in that it shows a decrease in the activation of the IRF pathway in response to compound treatment in both the WT and HERC4 KO cells. Additional validation could include demonstrating that AK59 reduces levels of IFN- β , CXCL10 and IL-6 in cGAMP stimulated THP-1 cells. They could also demonstrate that AK59 treatment decreases relative levels of phosphorylated TBK1.

They claim the data in Fig. 5 (b) and (c) support the hypothesis that AK59 does not reduce levels of known HERC4 substrates and is a specific degrader of STING. Those data show that levels of HERC4 in AK59 treated THP-1 cells, STING KO cells and cells overexpressing STING 141-341 are significantly reduced. The authors hypothesize that binding of AK59 to HERC4 stimulates auto-ubiquitination and subsequent degradation. They should demonstrate that experimentally or remove lines 247-255 and Fig. 5 (b) and (c). The authors should indicate in the figure legend that the data shown in 5(c) corresponds to a ubiquitin pull down and the data in 5(d) corresponds to a HERC4 pulldown.

The authors performed HERC4 pulldown experiments with cell overexpressing three STING variants, 141-341, 155-341 and K150R. Results reported in Extended Data Fig. 6 (a). They observed pull down of all three STING variants and claim that is evidence that the connector helix and K150 are essential for ubiquitination but not ternary complex formation. The flaw with that claim is that there are significant differences in the amount of STING protein pulled down between those three variants.

The most convincing data demonstrating the AK59 induced interaction between HERC4 and STING are the data reported in Fig. 5(e) from the NanoBIT luciferase reporter assay.

Materials and Methods

All reagents included in the materials and methods section should include both manufacturer and catalog number information.

Proteomics and analysis: The authors should include the number of cells that were used in the proteomics prep, the quantity of labeled peptides used and how they were detected, the LC gradient used to separate the peptides, and details on how the data analysis was performed (method for scaling the normalization, indications of any missing values in the data, how they dealt with any missing values and the method of imputation). The in-house R (v.3.6) script used to quantify proteins needs to be deposited in Github prior to publication and a citation to that code included in the manuscript.

Western blotting and co-immunoprecipitation: On line 431, the authors should indicate the concentration and volume of protein loaded.

Figures and Figure legends

Compound concentration used should be added to the legend for Extended Data Fig. 1(b).

The legend of Extended Data Fig. 1(d) should indicate what the red dots correspond to.

The legend of Extended Data Fig. 6(c) should include the time point at which the luminescence was read.

The label of the Y-axis of Figure 2(c) should indicate the concentration of AK59 used in the experiment.

Figure 2(d) should be edited so that the upper bands of lanes 1 and 2 are not cut off and it is clear that the same bands are not present in lanes 3-8.

To clarify what is represented in the plot, the Y-axis of Fig. 5(a) should be labeled “ Δ GFP signal STING155-341 - STING141-341” and the X-axis should be labeled “time after AK59 treatment”.

Appropriate use of statistics and treatment of uncertainties:

The authors should report EC50's and percent maximal inhibition for the dose response data shown in Extended Data Fig. 1 and Extended Data Fig. 5.

The authors used the appropriate statistical analyses of the CRISPR screening data (RSA, Q3) and for the identification of significantly enriched or depleted genes in the study groups (Euclidean distance between R,Q coordinates).

The authors should add to the figure legends of Fig. 2(f), 4(j), 5(e) Ext. Data Fig. 1(a), Ext. Data Fig. 5(c), Ext. Data Fig. 6(c) and (d) the p-values that correspond to the asterisk annotations.

The Y-axis of Extended Data Fig. 4 does not include the range of cell count values for each panel.

The X-axis of Extended Data Fig. 1 should indicate which concentration corresponds to the tick mark on the far right.

Conclusions:

As detailed in previous sections, there are several conclusions that are not supported by the data included. The conclusions in question are mostly supportive and could be bolstered by further experiments or removed without significantly weakening the central claims.

Suggested improvements:

The additional experiments that would significantly contribute to the work are experiments to demonstrate the functional effect of AK59 induced degradation of STING. Those could include demonstrating that AK59 reduces levels of IFN- β , CXCL10 and IL-6 in cGAMP stimulated THP-1 cells. They could also demonstrate that AK59 treatment decreases relative levels of phosphorylated TBK1.

References:

The authors should add references that support the statements on lines 33 and 35 and for the method cited on line 503. References 16-18 are not cited in the body of manuscript or in the figure legends. Reference 42 is a duplicate of reference 38.

Clarity and context:

The abstract is clear and accessible. The abstract, introduction and discussion sections are appropriate.

Inflammatory material: The manuscript does not contain any language that is inappropriate or potentially libelous.

March 12th, 2024

Dear reviewers,

Thank you very much for your thorough and constructive feedback. Please find below a point-by-point response of your comments. Our responses as well as the changes in the manuscript were highlighted in blue.

Reviewer #1 (Remarks to the Author):

Mutlu et al., describe development of a small molecule AK59 which acts as a HERC4 E3 ubiquitin ligase dependent degrader of STING. STING is a major element in innate immune system as part of cGAS-STING pathway and its inhibition of this pathway for autoimmune diseases have attracted considerable interest. The authors identified AK59 in a phenotypic screen for type 1 interferon inhibition and further validated it to downregulate STING. CRISPR screen identified HERC4 as a E3 ubiquitin ligase which could be responsible.

Overall study is well designed and identification of novel molecular glue degraders (while the mode of action remains to be fully elucidated) is of great interest to the TPD field.

There are few questions and clarifications that are need addressing prior to publication.

1. Could the authors comment on general toxicity of AK59 or QK50 in THP1 cells at 24 or 3 day timepoint - was that assessed?

We thank Reviewer 1 for pointing out this aspect. We acknowledge the lack of these initial experiments in the manuscript but we have evaluated the toxicity of AK59 over-time as one of the earliest experiments. We have now provided this data in Extended data figure 1a. We selected 16 hours of incubation of the compound which is the longest incubation time without seeing significant toxic as well as secondary effects AK59.

2. Paragraph in line 138-158. Mutation of lysine 150 to arginine. The authors identified the STING 141-155 region as critical for degradation, mutate lysine 150 to arginine, and then reason that that mutation is essential for ubiquitination of STING induced by AK59. Another, explanation would be that K150 is part of a 'degron' motif, or perhaps interacts with the AK59 directly, or is at the interface between HERC4 and STING. I would question the reasoning here - mutation could also reduce the binary binding affinity, or prevent ternary complex formation which would also affect ubiquitination.

We thank Reviewer 1 for emphasizing this point. Indeed, this was one of the questions in our minds therefore we performed the co-immunoprecipitation experiment using HERC4 antibody (extended data fig.7e). This data represents HERC4 interaction with various STING constructs (STING¹⁵⁵⁻³⁴¹, STING¹⁴¹⁻³⁴¹ and STING^{141-341_K150R}) in the presence of AK59 and bortezomib (to avoid protein amount difference due to degradation). Furthermore, we also introduced these same STING constructs into LgBiT tagging NanoBiT pair plasmid. We co-transfected these constructs with SmBiT-HERC4 expression construct to further understand the differences in ternary complex formation (Extended Data Fig. 7c). While all constructs showed luminescence signal in the presence of 10 μ M AK59, we have not seen any significant difference between constructs. Together, these data showed us that K150 residue is not essential for complex formation and might not be part of a potential degron.

3. Line 171/296 - I offer my skepticism on the identification of a lysine residue (from ubiquitination point of view) - please see comment above. That would need to be further validated - perhaps ternary complex, or compound binding is inhibited instead? NanoBit assay STING S150K:HERC4 would be able to answer this question.

We thank Reviewer 1 for raising this point. We have described the experiment on the comment above.

4. Line 268 and Extended data 6a. It appears that STING 155-341 pulls down with AK59, but longer construct which includes the helix STING 141-341 does not. In figure 2d - degradation data - it appears that STING 155-341 is not degraded by AK59, but 141-341 is. This appears to be contradictory. Why would STING-141-341 not interact with HERC4 upon compound treatment, but be potentially degraded?

We thank Reviewer 1 for raising this critical point. We observed pulldown of both STING constructs (155-341 and 141-341) with HERC4 in the presence of AK59 with various protein levels. We concluded that the protein level variation was due to degradation of STING141-341. In order to clarify this point, we rerun the pulldown experiment in the presence of bortezomib to avoid any bias due to degradation (extended data fig. 7e). In the new pulldown experiment, we provide the input STING levels and pulldown levels in the same blot (also check source file). Additionally, we run DMSO control samples for each STING construct expressed sample to further emphasize the compound independency of the interaction.

5. Did the authors consider testing the negative compound QK50 in a competition experiment with the AK59 in a nanobit assay? If successful, cells pre-treated with constant amount of AK59, with a dose response of QK50 could show competition, indicating that QK50 maintains binary binding to either STING or HERC4, but the quinoline in QK50 affects the formation of ternary complex specifically.

We followed the suggestion of Reviewer 1 and we tested this hypothesis. Due to toxicity of AK59 on cells (Extended Data Fig.1a) we could not construct the experiment as the Reviewer 1 suggested. Interestingly, QK50 does not show the similar toxicity effect on cells therefore we managed to setup the assay by inverting the suggested condition. STING¹⁴¹⁻³⁴¹-LgBiT:SmBiT-HERC4 expressing HEK293T cells were treated with QK50 while increasing doses of AK59 added on top (Extended data Fig.7b). As a control we have used equivalent DMSO addition on the cells prior to AK59 treatment. After 16 hours AK59 treatment, we saw that QK50 treatment competition did not cause a significant difference in the fold change of luminescence. This was a pivotal experiment showing that quinoline structure did not show binary affinity to either STING or HERC4 that might result in decreased complex formation.

Minor points:

Line 66. Please provide references for the expression level association with these cancers - is it over-expression or loss of ligase?

We thank Reviewer 1 for pointing this out. We have added the necessary references.

Fig 2c and d. or 5a The representation is rather confusing. It may be easier for the reader to have a downward trend to look at - synonymous with degradation. Perhaps a normalization to DMSO would allow for this?

We thank Reviewer 1 for raising this concern. We modify the presentation of the data as the reviewer suggest. In the revised manuscript, Fig.2a and Fig.2c are represented as delta GFP cell area/total cell area which is then normalized to DMSO (treatment-control). For Fig. 5a, as STING¹⁵¹⁻³⁴¹ was used as the control group, the figure modified as normalized delta GFP cell area/ total cell area normalized to STING¹⁵⁵⁻³⁴¹ construct (STING¹⁴¹⁻³⁴¹-STING¹⁵⁵⁻³⁴¹) with the downward trend to align with the degradation.

Fig 3d. Please consider labeling other top factors identified in the screen.

March 12th, 2024

We thank Reviewer 1 for this suggestion. However, we would prefer not to highlight all top factors on the main figure, to emphasize the main hits and to simplify the figure, since we already provide the whole list of hits in the Supplementary table S2. We have now re-organized the excel sheet to point out the top hits.

Reviewer #2 (Remarks to the Author):

STING is a protein associated with many inflammatory diseases. While the understanding of STING is far from complete, removal of STING by genetic approaches or inhibition of STING by small molecule inhibitors can ameliorate disease.

This study utilized several approaches and strategies to describe the identification of AK59-51B (abbreviated AK59) as a small molecule which induces STING degradation. Broadly speaking, the two main cell types utilized in this study are THP1 and HEK293. AK59 was first identified as a small molecule that type I interferon response upon exogenous DNA stimulation (this data was not shown). Different forms of STING were expressed in THP1 and HEK293 cells to assess the impact that AK59 has on the levels of STING. Furthermore, a CRISPR screen was employed in THP1 cells, enabling the identification of HERC4 as a mediator of STING degradation upon AK59 treatment. Several genetic knockout cell lines were used to complement the finding of the CRISPR screen. Interestingly, HERC4 and STING associate based on a luciferase complementation assay and immunoprecipitation experiments. The findings are interesting and relevant given the broad relevance of the STING pathway and targeted protein degradation.

The findings of the study are important to enable the discovery of new mechanisms to modulate STING levels for treatment of STING-associated diseases. The authors have used independent methods to generate data for their study. However, I highly recommend that fewer but well-designed experiments should be the focus of the authors to improve the quality of the manuscript, rather than executing too many different experiments with flaws in the quality of the data. In addition, a major concern is that most of the manuscript contains inaccurate interpretation of data either due to uncontrolled experimental design or inadequate information. It is still difficult to conclude whether AK59 acts as a molecular glue that generates a new interface between HERC4 and STING to then degrade STING. I recommend that the authors reconsider removing experiments and adding new experimental evidence. I believe the manuscript should be rejected from acceptance at Nature Communications unless the gaps and flaws of the experiments are considered.

We thank reviewer 2 for their critical revision of our manuscript. We hope that following additions have cleared the concerns of the reviewer.

The major comments for the manuscript are described here. Experiments and comments for specific sections are grouped by sub-numbering (1-1, 1-2, 1-3, 1-4).

Point 1: How well does AK59 degrade STING?

1-1: A time course of AK59 inducing STING degradation is needed along with concentration doses for both Figure 1 and/or extended Figure 1. This is important for 2 main reasons; (a) this is important for understanding the behavior of the compound and, (b) it is unclear why the current concentration and time is chosen for AK59. Treating for 16 h with 10 μ M of AK59 seems too long to induce STING degradation. It is unclear if this prolonged 16 h incubation will result in off-target cellular effects such as induction of cellular stresses or rewiring or cellular signaling, therefore causing STING degradation. This data will fit into figure 1 or extended figure 1.

We thank reviewer 2 for emphasizing this important point. In order to optimize the dose and incubation time of AK59, we have tested the compound effects on viability (Extended Data Fig. 1a); as well as

monitoring the dose response of AK59 on cGAS/STING pathway inhibition (Fig 1b, Extended Data Fig. 1b-d) and STING expression levels (Fig. 1c). We aimed to for the longest incubation time where we do not compromise on cellular viability. A concentration of 10 μ M was selected due to our observation on the stable STING degradation levels without compromising cell viability and adding complicating factors to the analysis.

Point 2: Does AK59 induce ubiquitination of STING?

2-1: The ubiquitin-based pulldowns can be improved to make the claim that STING is ubiquitinated in the presence of AK59. In figure 1g, 2b, 5c, it is not stated how long AK59 is present for. Please state this clearly as it influences interpretation. In figure 1g, lanes 3 and 4 do not appear different from lanes 1 and 2. I would expect 50 nM bortezomib to protect STING against AK59-induced degradation of STING, but this does not look to be the case compared to lanes 1 and 2 (see input blot). Furthermore, I would expect that ubiquitin pulldown to show more “ubiquitinated STING” (lane 4) upon AK59 treatment with bortezomib compared to AK59 without bortezomib (lane 2) (see IP: ubiquitin). Please provide an explanation for this. As AK59 degrades STING, less ubiquitinated STING should be available in the ubiquitin pulldown (lane 2) compared to if AK59 and bortezomib were both present (lane 4).

We thank reviewer 2 for this comment. We have now added the incubation time information on the indicated figures. Additionally, Fig 1h (previous Fig. 1g) and 2c (previous Fig. 2b) were exchanged to increase the blot quality and Figure 5c removed according to the reviewer 2's comments.

2-2: Following on from point 2-1, given the authors mostly utilize cell-based assays, it would be important to provide more evidence that AK59 drives degradation of STING through a ubiquitin signal of relevance (K48-linked polyubiquitin chains for example). This could be done by recombinant protein experiments or other types of cell-based experiments. One way to do this is to transfect the classical HA-ubiquitin constructs (WT, K48R, K63R) into either HEK293 or THP1 cells followed by 10 μ M AK59 treatment (or vehicle) in the presence or absence of 50 nM bortezomib. A pulldown with anti-HA followed by detecting for STING will provide evidence if AK59 drives degradation by STING through a K48-linked ubiquitin signal. This provides a mechanism of action for the claimed molecular glue. The authors can repeat the experiment in HERC4 KO cells if needed.

We thank reviewer 2 for this important point. We are aware the HA-ubiquitin introduction to the choice of cell line one uses is one of the common ways to show ubiquitination pulldowns. Unfortunately, this technique did not work efficiently on our hands. Transfection of the HA-ubiquitin construct together with other STING constructs brought too much variability in the expression and made it impossible to compare samples. Therefore, we performed all our experiments with an well-established and published ubiquitin pulldown kit from Enzo (Ubiquapture kit, <https://www.enzolifesciences.com/BML-UW8995A/ubiquapture-q-kit/>). This commercially available kit is a useful tool to detect mono-/multi-/poly-ubiquitin modified, lysine linkage independent ubiquitination patterns and it was sufficient for our hypothesis testing in the manuscript.

Point 3: Quality of data for STING constructs in Figure 2 requires major revisions.

3-1: The experiments in Figure 2 for the section “Lysine 150 of STING is essential for AK59-mediated STING degradation” is made up of a mix of experiments that do not seem to be controlled and contain inconsistent results from the rest of the manuscript. I would suggest removing this entire figure and entire section as it does not add value. If the authors wish to keep this section, I recommend changing the interpretation or redoing the experiments according to the other points below (3-2 to 3-11).

We thank reviewer 2 for their suggestion and we hope that our additions to the indicated figure would satisfy the reviewer.

3-2: In Figure 2a, please also include experiments for wild-type full-length STING tagged with GFP as this

best represents endogenous STING for consistency with Figure 1. Do the truncated versions / GFP-tagged versions get degraded faster than endogenous STING? This seems to be the case because maximal degradation occurs at 2 – 4 h for construct 1 (STING 141-341). Important to also relabel the legend to ensure authors note the construct number (construct #1, construct #4) AND whether the protein is GFP tagged or not.

We thank reviewer 2 for their suggestion. We have tested full-length STING side by side with our STING¹⁴¹⁻³⁴¹ and STING¹⁵⁵⁻³⁴¹ constructs in HEK293T cells (Fig.2c and Extended Data Fig. 2b). While we saw that full length STING similarly expressed in HEK293T cells as well as degraded (Fig.2c), we also saw different kinetics in live tracking of GFP tagged STING expressing HEK293T cells (Extended Data Fig. 2b). We believe this might be due to complexity of protein folding and expression of the transmembrane domain of STING which we also added to the main text of the manuscript. Thus, we believe that for a better comparison, we proceed with only cytosolic domain expressing constructs.

In order to make the legends clearer and easy to follow, we now removed the construct numbers and directly mentioned for each construct whether they are GFP tagged or not. This way the readers do not have to go back and forth to check construct numbers but easily access the information from the figure legend. We thank the reviewer for suggesting this helpful modification.

3-3: Why is the fraction of GFP+ cells ranging from 0.9 to 1.3 in the scale only? This seems to suggest the degradation is very partial in these assays. For example, 1.0 to 0.9 suggests a 10% reduction of GFP positive cells. Can the authors really claim that this STING 141-341-GFP is degraded by AK59?

We thank reviewer 2 for their critical observation. With these assays, we are looking at the fraction of cells that are expressing STING-GFP construct and among those the reduction of GFP signal between AK59 treatment to control (DMSO). Data represented in GFP + cell area/total cell area with a constant maximum and minimum intensities. Our broad maximum and minimum intensity cut-offs results in a stringent criteria on what if truly GFP negative (therefore STING degradation). Also, this cut-off would exclude slight changes in GFP signal which might be treatment independent and that would cause false positive results. Due to our stringent cutoffs, STING degradation might be underrepresented in the assay setup therefore we provide also end-point western blot results as an alternative method to assess our findings.

Additionally, these are transiently overexpressed constructs which have high and unnatural peak of expression for short period of time therefore we simply cannot compare the degradation levels with our THP1 model which has endogenous full length STING expression, nor with the effects in human PBMCs. We are aware that the assay window is narrow and maximum decrease of STING-GFP expression is low comparable to the THP as well PBMCs cell models. However, transient expression of various STING expression construct in HEK293T cells helped us investigate the functional domain of STING responsible in AK59-dependent degradation.

We believe our live tracking is align with our end-point western blot. Additionally, throughout the paper, the results from three different cell line models are comparable therefore we are confident in the conclusion of that AK59 brings about STING degradation.

3-4: Figure 2b requires input blot for STING and control with bortezomib. What is the basal level of STING 155-341, STING 141-341, STING 141-341 (K150R)? At this point, if the expression levels from the HEK293 transfection varies from construct to construct, authors cannot claim anything about relative ubiquitination levels or status between constructs, please revisit the statement "ubiquitination levels remained constant in either STING 155-341 or STING 141-341 K150R" (see line 155). Importantly no mention of treatment time is indicated. Is this a 16 h experiment as well? How is it possible to immunoprecipitate (pull down) ubiquitinated form of STING if AK59 is triggering substantial degradation of STING constructs? To confidently conclude that ubiquitinated form of STING is captured, authors need to redo the experiment in figure 2b with 50 nM bortezomib and a vehicle control to observe differences.

March 12th, 2024

We thank reviewer two for their comment. This experiment is now repeated in the presence of bortezomib treatment including the input levels of each STING expressing sample and the results are consistent with those initially reported.

3-5: Figure 2c data should be plotted as in Figure 2a. Replot as fraction of GFP+ cells (normalized to treatment start). In the current fold change graph, it is difficult to compare the findings to figure 2a. For example, in figure 2c, it is interpreted that STING 141-341 (construct #1) maximal degradation occurs at 12 – 16 h, whereas in figure 2a it isn't the case.

We thank reviewer 2 for this suggestion. We have now unified the plotting of the GFP live-tracking experiments as the Delta fold change (treatment-control). This way the trend of the plot is downwards representing the downregulation pattern of STING. Additionally, we have gone over each data carefully and included necessary replicates for consistent findings.

3-6: Figure 2a and 2c should be complemented with cell confluence data in the IncuCyte. This is to ensure cell numbers are not changing across time during the experiment. Please provide this data. In the current form, figure 2a and 2c can easily be affected by variability of IncuCyte plating density.

We thank reviewer 2 for their suggestion. We agree with the reviewer on the importance of the well confluency. We did not provide the confluency data of each experiment because this has been already taken into consideration in the analysis of the GFP therefore all the data that has been already shown in the manuscript was already normalized to total cell area. We apologize if this information was not clearly presented. We now added the detailed information on the methodology as well as in the necessary figures.

3-7: Figure 2a and 2c should be complemented with GFP intensity measurements. Rather than just counting GFP positivity (yes GFP present, or no GFP is absent), it is important to report GFP mean intensity from the IncuCyte as it resolves high GFP to low GFP signal. This gives another important layer of information on how fast acting AK59 is in terms degrading STING-GFP constructs.

We thank reviewer 2. As we previously explain in point 3-3, we have presented the live GFP tracking data as GFP+ cells area/total cells are with a consistent cut-off throughout the replicates. We have now added channel intensity cut-offs in the methods section. The analysis classifies GFP positivity according to the minimum and maximum cut off defined in the Incucyte software and normalizes it to first total cell area and then to the treatment start time point. Presenting data in GFP intensity would be more prone to smaller changes in GFP signal which might be independent from the treatment.

3-8: In figure 2, it is not clear how many times the experiments were done. It seems the experiments in the entire manuscript were only done once. Please state clearly with the following statements: (a) "Western blot is representative of 3 independent experiments", (b) "The IncuCyte experiments were carried out in 3 replicates and repeated in 3 independent experiments, the graph shows mean \pm standard deviation from the replicates of 1 experiment". This comment is highlighted in in minor comments as well as a general note for methods reporting in figures.

We thank reviewer 2 for their careful observation. We have now indicated the replicate information in each figure legend and added replicates of the experiment wherever it was necessary.

3-9: Are the constructs express to the same extent or variably? If there is too much STING expressed, can AK59 degrade it? In figure 2, it is essential to provide an assessment of STING protein expression by Western blotting or mean GFP fluorescence intensity by FACS. A quantification of relative protein levels by Western blot or FACS will give an idea whether the STING constructs express similarly. This is important

to report variability in experiments and how much impact does STING protein levels have on efficacy of AK59 to degrade STING. Please address this technical consideration point by experiments.

We thank reviewer 2 for this point. Construct expression levels can be seen in Fig.2c, 2e as well as Extended data Fig.2c.

3-10: STING variants are claimed to be “partially degraded” in figure 2d and 2c. In figure 2d, how much degradation is occurring? Provide quantification of the Western blot in figure 2d this reduction in STING V155M and N154S by AK59 appears to be variation in loading. Also, in figure 2c, the fold change from 0 to 0.2 for STING V155M and N154S could be due to variability in cell density or cell growth. Please plot additional data as in points 3-6 and 3-7 above to provide cell confluence and GFP intensity data as well.

We thank reviewer 2 for re-emphasizing this point. We have now repeated Extended Data Fig.2c (previous Fig.2d) and also provided the quantification of the bands. As we previously described, GFP tracking experiments are normalized to total cell area therefore the effect reported on SAVI mutants are independent of cell density or growth.

3-11: Overall, figure 2 requires substantial rework and additional experiments to support the title “lysine 150 of STING is essential for AK59-mediated STING degradation”. The most fundamental experiment is to express STING full length (WT) and STING full-length (K150R) in HEK293T followed by 10 μ M AK59 or vehicle treatment in the presence of bortezomib or a vehicle control. This experiment alone could support the title well.

We thank reviewer 2 for their suggestion. We have now provided the full-length STING expression results in Fig. 2c and Extended Data Fig.2b. Due to significant difference in cell growth possibly due to difficulties in expression (with STING being an ER associated protein), we decided to proceed using cytosolic domain expressing constructs, as these gave more consistent and reproducible results. We hope that these results would be sufficient for reviewer’s suggestion.

Point 4: The role of HERC4 in AK59-induced degradation of STING

4-1: The authors carried out a successful genetic based screen to identify regulators of AK59-induced degradation of STING, testing ubiquitin-related proteins from their top hits. HERC4 knockout provided only partial protection against AK59-induced degradation of STING in figure 4, why might this occur? Please provide an explanation in the discussion or results. Could it be possible that the other top hits (FASTKD5, PMAIP1, UQCRFS1) have a greater role in AK59-induced degradation of STING?

We thank reviewer 2 for their input. We have followed up on top hits of the CRISPR/Cas9 screen. While we highlight HERC4, UBA6 and UBA5 due to their potential interaction as well as the validation, we believe that AK59 could have secondary effects as well as potential additional partners to the HERC4-AK59-STING interaction. We have now extended our discussion on this point on the manuscript.

4-2: Is the E3 ligase activity of HERC4 needed for AK59-induced degradation of STING? This will be important to address. Otherwise, please remove the phrasing “E3 ligase HERC4” throughout the text (line 237, line 246 and anywhere else in the manuscript where needed). It’s not clear at this point if HERC4 E3 ligase function is directly linked to degradation of STING by AK59. It would be informative to test if HERC4 WT compared to catalytic dead mutant overexpression will be favoring AK59-induced degradation of STING.

We thank reviewer 2 for their critical thinking. We tested HERC4 catalytic dead mutant on our Nanobit complementation assay setup where we transfected cells with SmBiT-HERC4^{C1025A} and STING¹⁴¹⁻³⁴¹-LgBiT expression vectors and compared the fold difference in luminescence upon AK59 treatment. We saw that the catalytic dead mutant showed a trend of increased fold change in luminescence (Extended Data Fig. 7d). First, this provides further evidence that the ternary complex formation is independent from

the catalytic site of HERC4. Second, introduction of the mutation stabilized of the luminescence signal indicating that degradation of the complex through HERC4 E3 ligase function may be occurring, reducing the apparent signal.

4-3: Figure 5a contains multiple flaws including insufficient data, contradictory data, unclear data. Firstly, figure 5a needs to have additional plots (confluence graph, GFP intensity). as it faces the same issues with earlier IncuCyte experiments. Next in this experiment, it is not clear which STING-GFP construct was used even though authors refer to extended figure 2. Please specify in materials and methods section, figure legend and results section. This is major information that needs reporting. This will affect the interpretation of data and will determine whether further experiments are needed. The other flaw to address is whether this is from one experiment or merged data from different experiments. The current data shows that if the standard deviation bars were plotted at both ends of the mean, both data for the black (control sgRNA) and red lines (HERC4 sgRNA) would overlap. This will then mean there's no change between ctrl sgRNA and HERC4 sgRNA.

We thank reviewer 2 for their comment. As previously discussed, we presented our Incucyte data as delta (treatment-control) GFP+ cell area/total cell area normalized to treatment start point. We have also now added the necessary explanation to Fig.5b legend as well as corresponding method section. As previously mentioned, we placed broad cut-offs for GFP positivity for a stringent "GFP negative" selection. This way we are striving to make sure that the results presented are not false positive fluctuations of GFP signal independent from treatment condition.

We are aware that the current data has higher error bars in longer incubation times. This is also aligned with our findings that there is partial rescue of AK59-dependent STING degradation in HERC4 knockout lines as well as potential secondary effects of the compound. We acknowledge these findings in our results and discuss it on the manuscript. We are aiming to present the data in highest quality achievable within the given circumstances and we would like to emphasize that the data presented in our manuscript has been meticulously collected and analyzed to the best of our abilities, taking into consideration the limitations and constraints inherent to the study design.

4-4: Due to the insufficient data that HERC4 is mediating AK59-induced degradation of STING, authors need to generate more data. For example, this could involve HERC4 knockout in THP1 and HEK293T cells, followed by AK59 treatment and assessment of endogenous STING by Western blotting. A control such as 50 nM bortezomib is needed. It is important to also repeat such experiment with the STING constructs, specifically construct #1 (STING 141-341-GFP), construct #5 (STING 141-341). Can HERC4 knockout rescue AK59-induced degradation of STING and how does this compare to bortezomib treatment to block degradation?

We thank reviewer 2 for this point. We already shown HERC4 KO effect on STING¹⁴¹⁻³⁴¹-GFP and STING¹⁵¹⁻³⁴¹-GFP degradation in HEK293-JumpIN-Cas9 cells (iFig.5b) Additional to this information, we have now provided the data showing the change in total proteome on THP1 HERC4 KO cells treated with 10 μ M AK59 compared DMSO control group (Extended Data Fig. 6b). In this data, while we are seeing the STING remaining stable in the presence of AK59, we are also observing an effect of AK59, in the absence of HERC4, on rest of the proteome. We believe this is important source of information to show HERC4-AK59 dependent STING regulation in unbiased manner but also to acknowledge compound's potential targets, mechanism of action as well as secondary effects. Additionally, we also performed the same experimental approach on STING knockout THP1 cells (Extended Fig. 6c). We believe these results will be beneficial for follow up work on identifying novel targets of AK59 in both HERC4-dependent and independent mechanisms.

4-5: It is unclear why data in figure 5b and figure 5c currently sit in the main figures as they do not add further understanding into how AK59 works as a potential molecular glue. It is however important to

improve figure 5b as the quality of the data is questionable. Firstly, is reproducibility, in figure 5b, this is the first time that STING is completely absent with AK59 treatment, is this real reproducible result? I would suggest moving figure 5b to extended data and repeating the data to provide more repeats, running multiple experiments on a single experiment. This is important to make the claim AK59 degrades STING partially or completely. Second is the finding that HERC4 disappears with AK59 treatment. It is important for this figure because the observation that HERC4 is degraded by AK59 is novel. One major issue is that the HERC4 band is noticeably below 115 kDa whereas in all other figures HERC4 is at least 115 kDa which corresponds to approximately the right size.

We thank reviewer 2 for raising these points. We have modified the main figure 5 considering the reviewer comments and removed Fig 5c to concentrate the reader's attention to the significant finding. We repeated HERC4 co-immunoprecipitation assay in the presence of 10 μ M AK59 and 50 nM bortezomib to avoid changes in protein levels due to degradation (Fig.5c).

HERC4 degradation upon AK59 exposure was a consistent observation throughout this project. We believe this is due to an auto-ubiquitination mechanism which has been defined before for HECT-type ligases¹⁻³. While it is remarkable to see the compound potentially degrading the E3 ligase it is working with, it is also significant to point out how much this increases the complexity of the mechanism of action within cells.

Regarding the size of the HERC4 band, we have consistently observed the HERC4 band (~119kDa) close to 115kDa ladder band which can also be seen in source data file. We realized formatting mistake on the Fig. 5c and corrected it. We thank reviewer 2 for pointing this mistake out. Original blots can be also seen in source file.

Regarding the STING degradation level, we believe that our findings are consistent. New additional experiments in extended data Fig 1c and 1h are in line with Fig 5c findings which represents significant decrease in STING expression.

4-6: Figure 4c has the same issues as earlier immunoprecipitation experiments in figures 1g and 2b. There are multiple concerns and questions for this experiment that require further explanation or experiments. (1) An input blot for STING and HERC4 is essential here to demonstrate that AK59 is functioning as previous figures. (2) Also, the number of independent experiments must be reported. (3) How long is this experiment with the AK59 treatment? (4) Given AK59 triggers STING ubiquitination degradation, how is it possible to still immunoprecipitate (pull down) substantial amount of ubiquitinated STING (lane 3 and lane 4, STING blot)? (5) The experiment needs to be repeated as there is obvious differences in loading control (tubulin) which could lead to inaccurate interpretation of the immunoprecipitation. For example, authors claim that less of Smoothened was co-immunoprecipitated (lane 4) but this could be less starting material, evident by lower tubulin loading control.

We thank reviewer 2 for this point. We have now provided the immunoprecipitation experiments in the presence of bortezomib treatment to avoid protein amount changes due to treatment (Fig. 5c, Extended Data Fig. 7e). We wanted to point out again that HEK293T model has transient transfection of various STING constructs where this causes short-term unnaturally high expressions of the protein. Therefore, it would be possible to still detect STING expression in the overexpression systems even though the protein is being targeted for degraded.

4-7: Smoothened remains unchanged despite HERC4 loss with AK59. Smoothened is supposedly degraded by HERC4. In figure 5b and 5c, Smoothened loss is not observed with AK59 treatment. Why might this be the case? In figure 5 and extended data figure 6, the authors are trying to demonstrate that AK59 induces HERC4-STING complex formation and that AK59 interferes with HERC4-Smoothened complex formation. Thus far, the data have not been convincing. This is confounded by the fact that HERC4 itself is loss with AK59 treatment. One way to demonstrate this is to generate SmBiT-HERC4 and

Smoothened-LgBiT constructs and express them in HEK293T cells. If AK59 generates a new interface to favor STING-HERC4 interaction, it would be expected that AK59 would reduce Smoothened-LgBiT/SmBiT-HERC4 interaction, which can be measured by luminescence. More experiments are needed to support the model in figure 5f. The model in figure 5f “naïve state” shows that HERC4 degrades Smoothened even though this is not the case in the hands of the authors.

We thank reviewer 2 for this suggestion. We performed the suggested experiment however due to technical issues we could not provide additional data to further prove our findings. We acknowledge the reviewers concerns greatly therefore, we excluded this conclusion from the results and modified the manuscript accordingly.

4-8: Do STING truncated constructs interact with HERC4 upon 10 μ M AK59 treatment? Authors state that the STING constructs interact with HERC4 (lines 268 to 272) with AK59 treatment, but this is not shown convincingly. Extended data figure 6 should be repeated with proper controls, including DMSO controls. Authors also cannot make the claim in lines 271 and 272, that the connector helix and lysine 150 are essential for ubiquitination but not for formation of ternary complex in presence of AK59. To make such claim would require direct ternary complex formation experiments or mutagenesis experiments that abrogate HERC4-STING interaction.

We thank reviewer 2 for pointing out this suggestion. In our previous version, we provided HERC4 pulldown on various STING constructs. We have now advanced this result by adding DMSO control side by side with 10 μ M AK59 treatment, all samples in the presence of prior 50nM bortezomib treatment in order to maximize the pulldown content (Extended Data Fig. 7e). Together with this, we have now provided the NanoBiT complementation assay with various STING construct (141-341, 155-341 as well as K150R mutant) in order to question whether truncated versions of STING forming ternary complex with HERC4 and AK59 (Extended Data Fig. 7f). Both HERC4 pulldown as well as NanoBiT complementation assay results indicates that all the truncated constructs of cytosolic domain of STING resulted in the direct interaction to HERC4 in the presence of AK59, including the K150R mutant.

4-9: Direct ternary complex formation between HERC4 and STING remains questionable. The authors have used a complementation luciferase assay, a conventional method to show protein-protein interaction. The authors claimed the connector helix of STING is required for degradation by AK59. Intriguingly, STING 155-341 (lacking connector helix) still interacts with HERC4 (extended data figure 6a). What is the minimal region required for STING-HERC4 interaction with AK59 treatment? One way to address this is by using the SmBiT-LgBiT assay but introducing mutations/truncations along the STING construct 141-341 in the 4 key helical regions (157-189, 263-274, 281-301, 325-338). Given cGAMP co-treatment does not interfere with STING-LgBiT/SmBiT-HERC4, the ligand binding pocket could be dismissed.

We thank reviewer 2 for this suggestion. We also have thought about creating further truncated STING constructs to test either with HERC4 pulldowns or NanobiT complementation assays to narrow down to a degron motif. But once you start having shorter constructs you are compromising on the correct folding of the protein which could lead to degradation independent from treatment or even no expression. In the case of NanobiT complementation assay, truncating one of the proteins in ternary complex could further compromise the proximity of tags which will lead to false negative results. We acknowledge that the most elegant and straight forward way to show direct binding should be done by structural or biochemical approaches. However, our numerous attempts to isolate HERC4 protein have failed. We have also tried to biochemically show A59 and STING binding without HERC4 component which also failed (data not shown) however this is an expected phenomenon in the context of MGD meaning three components of the complex is necessary for binding affinity to surpass.

4-10: Mechanism of action and specificity of AK59 in cells and on HERC4 requires further clarity. While it is tempting to speculate AK59 is a molecular glue making STING a neo-substrate for HERC4, AK59 itself

also decreases the levels of other proteins (figure 1d proteomics data). Moreover, HERC4 itself disappears with AK59 treatment. In the schematic proposed (figure 5f), AK59 acts on an interface within HERC4. It also acts between HERC4 and STING (figure 5f). This will be important to confirm and test given the proposed mechanism is that AK59 is a molecular glue. The specificity of AK59 is questionable. It could also be that AK59 creates a new interface between HERC4 and many other proteins (multiple neosubstrates), not just STING, which would argue against the specificity as suggested in the title of the manuscript "A novel HERC4-dependent glue degrader targeting STING". A proposed experiment is to revisit some of the proteins from proteomics in figure 1d and assess by Western blot whether some of the decreased proteins are also degraded by AK59. Intriguingly HERC4 is in the proteomics data of figure 1d with log₂fc of -0.72 and q-value of 1.63E-08 but this was not reported. Alternatively, a structural / biochemical approach to show ligand binding of AK59 onto either HERC4, STING or both will be needed. Otherwise, the title of the manuscript should be rephrased to "A degrader targeting both STING and HERC4".

We thank reviewer 2 for this comment. We are aware that HERC4 levels are also decreased in the presence of AK59 which we already shown this finding experimentally in Extended Data Fig. 6a. Together with proteomics data from THP1 wildtype cells, we have discussed our findings and hypothesized on a potential compound-mediated disruption of an intramolecular auto-inhibitory state of HERC4. Intramolecular auto-inhibitory state of HECT domain containing E3 ligases is a phenomenon that previously described^{1,2}. During revision of this article others have reported the crystal structure of HACE1, a HECT-type E3 ligase where they describe its' substrate-independent dimerization inducing auto-inhibition. In light of our data, we are confident that HERC4 might have a similar structure and mechanism. While we can only hypothesize that HERC4 might function similarly, we agree with reviewer 2 on the importance of structural and biochemical experiments to further support our findings. As we mentioned on point 4-9, our many attempts in expressing full-length HERC4 was unfortunately unsuccessful.

We agree with the reviewer on the short comings of the AK59 and therefore modifying the title of the manuscript, but we believe we are reporting more than "A degrader targeting both STING and HERC4". Our data indicates direct binding of HERC4 and STING only in the presence of AK59 (pulldown and NanoBiT complementation assays) and loss of HERC4 results in rescue of compound-dependent STING degradation to an extent (HERC4 knockout validation and proteomics data). Considering also the suggestion from the reviewer, we have now changed the title of our manuscript to "Small molecule induced STING degradation facilitated by the HECT ligase HERC4".

Point 5: Experiments needing clarity

5-1: Figure 1g, 2b, 5c have the ubiquitin band in the immunoprecipitation blots being inconsistent. Is the band size correct? Is the ubiquitin band below 30 kDa, 25 kDa or 15 kDa?

We thank the reviewer 2 for their careful observation. We have now corrected the protein ladder labeling and provided the full blots in the source file.

5-2: Figure 5d shows that AK59 does not degrade STING (input blot, lane 2 compared to lane 3). Why is this happening? HERC4 in the input blot also needs to be shown. It is also unknown how many times this experiment has been repeated.

We thank reviewer 2 for their observation. According to the suggestions below, this experiment is now performed in the presence of bortezomib treatment and the representative blot of three experimental replicates were shown in the Fig 5c. Replicate information is added to the legend of the corresponding figure.

The minor comments with data and/or data presentation are described below.

Point 1: Figure legend and figure labelling

1-1: In all figure legends, it is not clear how many times the experiments were done. Now, it seems the experiments in the entire manuscript were only done once. Please state clearly with the following statements: (a) "Western blot is representative of 3 independent experiments", (b) "The luciferase / IncuCyte / FACS experiments were carried out in 3 replicates and repeated in 3 independent experiments, the graph shows mean \pm standard deviation from the replicates".

We thank reviewer 2 for this comment. We have now added the biological replicate information of each experiment in the corresponding figure legends.

1-2: In extended data figure 4, labelling for the legends has errors. I believe the labels need to correct to include DMSO for the last 4 conditions. All the treatments are currently AK59 and duplicated for some it seems.

We thank reviewer 2 for pointing out this mistake. We have corrected it.

Reviewer #3 (Remarks to the Author):

Key results:

The authors report the first molecular glue degrader that engages the HERC4 ligase. They also report the first molecular glue targeting STING, a high-value therapeutic target. The authors used a combination of proteomics, genomics and cellular assays to identify and validate the mechanism of action of their compound.

Validity:

Additional validation of the functional effect of compound-induced STING degradation is important. The manuscript includes data in Extended Data Figure 5 that shows a decrease in the activation of the IRF pathway in response to compound treatment; however, the data included also shows that there is a response in the HERC4 KO cells as well. The authors don't provide an explanation as to why there is a response in the HERC4 KO cells. Other publications reporting STING PROTACS, such as Liu, et al. J. Med. Chem 2022, demonstrate that their compound reduces levels of IFN- β , CXCL10 and IL-6 in cGAMP stimulated THP-1 cells. They also demonstrate that compound treatment decreases relative levels of phosphorylated TBK1. The authors should perform additional experiments that show similar effects on cytokine release and downstream target modulation. Additionally, those effects should be unique to wild-type THP-1 cells relative to the HERC4 cells or they should try to understand why there is a response in both wild type and HERC4 KO cells.

We thank Reviewer 3 for this important point. We are aware that we do see to a certain degree AK59-dependent decrease IRF pathway activation even in HERC4 knockout lines. This finding is parallel with the partial rescue of STING levels upon HERC4 knockout (Fig 4e and 4f.). We highlighted these findings and further discussed it in discussion.

Additionally, we have further investigated AK59 (and QK50) -dependent inhibition of cGAS/STING pathway downstream activation by detecting phosphorylation of IRF3, TBK1 as well as measuring CXCL10 and IFN beta levels from cell culture. We have demonstrated effective inhibition of phosphorylation of IRF3 and TBK1 upon AK59 but not in QK50 treated THP1 cells (Extended Data Fig. 1b). CXCL10 and IFN beta levels were also reported in AK59 or QK50 treated cGAMP-stimulated Dual-THP1 cells (Extended Data Fig. 1c and 1d). Furthermore, the same experiments also performed in HERC4 knockout background (Extended Data Fig. 5c-e). These findings were aligned with the IRF pathway reporter assay findings which we reported in the first version of the manuscript. As a summary,

AK59 treatment showed significant inhibition in cGAMP stimulation-driven IRF pathway activation. Again, we pointed out the point that HERC4 loss does not fully restore AK59-dependent IRF pathway inhibition and we acknowledge the fact that there might be secondary factors playing in AK59 mechanism.

Originality and significance:

The field of targeted protein degradation is very active and there are many scientific research groups working to identify additional E3 ligases that can be therapeutically leveraged. The authors report the first targeted molecular glue degrader that engages the HERC4 E3 ligase. The validation of this E3 ligase opens up new avenues for researchers in the field and is of significant interest. STING is a highly validated therapeutic target that has been demonstrated to be involved in multiple inflammatory and autoimmune disorders. While there is research reported in the literature identifying inhibitors of the enzymatic activity of STING, there are only a few reports of alternative mechanisms of preventing STING activity in the cell. There is a 2022 publication reporting a PROTAC molecule that degrades STING via the CRBN E3 ligase; however, this manuscript reports the first molecular glue targeting STING. These findings are of significant interest to the field because of the novel E3 ligase targeted (HERC4), the more desirable physico-chemical properties of the molecule discovered (AK59-51TB) relative to the PROTAC that targets CRBN and the ability of AK59 to induce degradation of the SAVI mutants of STING.

We thank Reviewer 3 for their acknowledgement of the significance of our work.

Data & methodology:

-Discovery of AK59

The authors should include a detailed description of the screening methodology that was used to identify AK59. Currently there is a single sentence on lines 92-94 that gives no detail and no data is included. It would increase the significance and impact to the field if the authors included a description of the screening campaign with the screening assay used, the number of compounds screened, what the readout of the assay was, hit calling criteria, hit rate, hit confirmation method used and hit confirmation rate. The authors should also include a figure in the extended data section that illustrates the screening funnel and a scatterplot of the primary assay data showing the signal of all compounds tested, the threshold of hit calling and indication of where the hit related to AK59 is in that scatter plot.

We thank Reviewer 3 for their suggestion. There are number of reasons why we did not provide additional information on the initial screen which led us to identify AK59. One of the major reasons we have not explained the discovery of the compound in detail was because it contains proprietary data and follow ups are still ongoing for some of the identified hits. We do not think that descriptions of these latter have any relevance to AK59. In particular these hits have a completely different mode of actions, which we are still trying to fully clarify. This report only focuses on the characterization of the MoA of AK59 rather than its' discovery.

To clarify the identification of AK59, we have now added an explanation of our initial efforts. In short, initial screen was a cell-based assay that was designed to identify compounds that would mediate dimerization of STING. During the screening campaign, different hits were identified that functions as agonists but also antagonists for STING activation. Starting with the initial screen (1.2×10^6 compounds), we have performed a counter screen (171 compounds), IRF reporter assay screen of potential hits (11 compounds), further optimization of the compound and characterization in biophysical assays. AK59 and QK50 was identified within the structure-activity relationship (SAR) assays around hit compounds from the initial screen.

As you can appreciate, the research behind identification of AK59 (and QK50) is more than a single screen and it is beyond the scope this manuscript. We are aware that our initial explanation in the

March 12th, 2024

manuscript might be underwhelming for scientific community. We hope that the current version is explanatory and satisfying to the scientific audience.

-Role of Lys150 in STING degradation

The western blot data shown in Fig. 2b indicates that there is still some level of ubiquitination of the K150R mutant of STING, although at a reduced level relative to WT. The authors should provide a hypothesis as to why the K150R mutant is ubiquitinated.

We thank Reviewer 3 for their critical observation. We have observed basal level of ubiquitination of all STING constructs that we transiently overexpressed in HEK293T cells. We believe this might be due to unnatural peak expression of the protein due to transfection. Furthermore, we know there are additional lysine sites on STING (including our short constructs) that enable the formed protein to be ubiquitinated. In the light of all this information, we concluded that it is not uncommon to see residual ubiquitination of the STING constructs independent from AK59 treatment. We have now added further explanation to the manuscript.

Role of HERC4, UBA5 and UBA6 in compound induced STING degradation.

The authors performed a genome wide CRISPR screen to identify genes that were essential for AK59 induced STING degradation. They developed a FACS-based assay for monitoring the level of STING in cells. The resulting data convincingly show the essential role of HERC4 and UBA5 in controlling STING levels. The CRISPR data is less convincing for UBA6. The authors report that they were able to identify interactions between HERC4, UBA5 and UBA6 reported in the STRING database and cite that supportive evidence of the role they play. My search of the database did not find a reported interaction between any of the three proteins. My search included alternative names for HERC4 (KIAA1593), UBA5 (THIFP1, UFM-activating enzyme, UBE1DC1) and UBA6 (MOP-4, E1-E2, UBE1L2). The authors should include in the methods section a description of the search parameters used in the STRING database that will demonstrate those interactions. I was able to confirm that there was no interaction between STING and either HERC4, UBA5 or UBA6 reported in the STRING database.

We thank Reviewer 3 for their careful observation. We are aware of partial rescue of AK59-dependent STING degradation by loss of HERC4, UBA5 and UBA6 at a certain extend. We made this point further clear in the results as well as in discussion.

Regarding STRING db. results, it is unfortunate coincidence that STRING db. uploaded their newest version (Version 12) in July 26th 2023, during the revision process of our manuscripts. We wanted to point out that we also shared the version information of the database in order to avoid such misunderstandings. You can still visualize the interactome we shared in this manuscript by running the previous version of STRING (Version 11.5, <https://version-11-5.string-db.org/>). The interaction between UBA6-HERC4 relies on co-expression in *Homo sapiens* as well as *Bos taurus*.

GENE COEXPRESSION

Coexpression observed in your query organism (Homo sapiens):

UBA6 - Ubiquitin-like modifier-activating enzyme 6; Activates ubiquitin by first adenylating its C-terminal glycine residue with ATP, and thereafter linking this residue to the side chain of a cysteine residue in E1, yielding a ubiquitin- E1 thioester and free AMP. Specific for ubiquitin, does not activate ubiquitin-like peptides. Differs from UBE1 in its specificity for substrate E2 charging. Does not charge cell cycle E2s, such as CDC34. Essential for embryonic development. Required for UBD/FAT10 conjugation. Isoform 2 may play a key role in ubiquitin system and may influence spermatogenesisi [...]
HERC4 - Hect and rld domain containing e3 ubiquitin protein ligase 4; Probable E3 ubiquitin-protein ligase HERC4; Probable E3 ubiquitin-protein ligase involved in either protein trafficking or in the distribution of cellular structures. Required for spermatozoon maturation and fertility, and for the removal of the cytoplasmic droplet of the spermatozoon. E3 ubiquitin-protein ligases accept ubiquitin from an E2 ubiquitin- conjugating enzyme in the form of a thioester and then directly transfer it to targeted substrates
RNA coexpression score 0.100

Coexpression of orthologs in other organisms:

Bos taurus:

UBA2 - Bos taurus ubiquitin-like modifier activating enzyme 2 (UBA2), mRNA
TRIP12 - E3 ubiquitin-protein ligase TRIP12; E3 ubiquitin-protein ligase involved in ubiquitin fusion degradation (UFD) pathway and regulation of DNA repair. Part of the ubiquitin fusion degradation (UFD) pathway, a process that mediates ubiquitination of protein at their N-terminus, regardless of the presence of lysine residues in target proteins. In normal cells, mediates ubiquitination and degradation of isoform p19ARF/ARF of CDKN2A, a lysine-less tumor suppressor required for p53/TP53 activation under oncogenic stress. In cancer cells, however, isoform p19ARF/ARF and TRIP12 are located in [...] coexpression score 0.227
UBA2 - Bos taurus ubiquitin-like modifier activating enzyme 2 (UBA2), mRNA
UBE3A - Ubiquitin-protein ligase e3 a; Bos taurus ubiquitin protein ligase E3A (UBE3A), mRNA coexpression score 0.178
UBA5 - Ubiquitin-like modifier-activating enzyme 5; E1-like enzyme which activates UFM1 and SUMO2
UBE3A - Ubiquitin-protein ligase e3 a; Bos taurus ubiquitin protein ligase E3A (UBE3A), mRNA coexpression score 0.110
UBA6 - Ubiquitin-activating enzyme e1-like protein 2; Bos taurus ubiquitin-like modifier activating enzyme 6 (UBA6), mRNA
ITCH - Bos taurus itchy E3 ubiquitin protein ligase homolog (mouse) (ITCH), mRNA coexpression score 0.081
UBA6 - Ubiquitin-activating enzyme e1-like protein 2; Bos taurus ubiquitin-like modifier activating enzyme 6 (UBA6), mRNA
HERC4 - HECT and RLD domain containing E3 ubiquitin protein ligase 4 coexpression score 0.173

To further investigate this, we have contacted STRING db help center to clarify the change in the database. STRING db. explained as *“The main difference is in the text-mining transferred evidence coming from other organism than human. which has an additional score of 0.2 Overall we reduced the score of this interaction from 0.4 to 0.25. 0.4 is the default cut-off used for visualization (you can change it in the settings tab) and the reason you don't see it in v12, while in v11.5 is still there at the default settings.”*

We have now specified the cut-off in our method section.

The single gene KO studies provide convincing evidence that HERC4, UBA5 and UBA6 have a role in the level of STING in AK59 treated cells. The western blot data in Fig. 4(h) for sgRNA1 shows the band for the protein loading control, Vinculin, is significantly weaker than for the control and sgRNA2 samples. That experiment should be repeated so that the levels of Vinculin are the same for all three samples.

We thank Reviewer 3 for their watchful comment. We have realized that we made a mistake and did not upload the most recent version of Fig. 4. We have now provided the correct Western blot with comparable VINCULIN levels in UBA5 knockout and control cell lines (Fig. 4d).

-AK59 is a MGD that induces an interaction between STING and HERC4

As mentioned previously, additional validation of the functional effect of compound-induced STING degradation is needed. Data reported in Extended Data Figure 5 is ambiguous in that it shows a decrease in the activation of the IRF pathway in response to compound treatment in both the WT and HERC4 KO cells. Additional validation could include demonstrating that AK59 reduces levels of IFN- β , CXCL10 and IL-6 in cGAMP stimulated THP-1 cells. They could also demonstrate that AK59 treatment decreases relative levels of phosphorylated TBK1.

We thank Reviewer 3 for this point raised. We have run the indicated experiment and provided the data in Extended Data Fig. 5c-e. (previously described above, under "Validity" point).

They claim the data in Fig. 5 (b) and (c) support the hypothesis that AK59 does not reduce levels of known HERC4 substrates and is a specific degrader of STING. Those data show that levels of HERC4 in AK59 treated THP-1 cells, STING KO cells and cells overexpressing STING 141-341 are significantly reduced. The authors hypothesize that binding of AK59 to HERC4 stimulates auto-ubiquitination and subsequent degradation. They should demonstrate that experimentally or remove lines 247-255 and Fig. 5 (b) and (c). The authors should indicate in the figure legend that the data shown in 5(c) corresponds to a ubiquitin pull down and the data in 5(d) corresponds to a HERC4 pulldown.

We thank Reviewer 3 for this critical point. We understand that we should clarify this part of the manuscript and continue with more solid reasonings.

Decrease of HERC4 levels upon AK59 was a consistent observation we saw (in Western blots as well as in pulldowns). However, we also acknowledge that ubiquitin pulldown in STING¹⁴¹⁻³⁴¹ overexpressing HEK293T cells lines proving this point fully therefore we are removing this data from the figure. In the revised manuscript, we change the phrasing of our findings. We hypothesize that HERC4 protein level changes might be due to a compound-mediated potential disruption of an intramolecular auto-inhibitory state of the HECT E3 ligase which might be leading to its degradation. This mechanism has recently reported by During¹ and colleagues on another HECT-type E3 ligase, HACE1. They showed auto-ubiquitination of HACE1, independent of the target leading to auto-inhibition. We believe that HERC4 could have a similar mechanism. We have now added this reference on our manuscript (in discussion) as well.

Additionally, we have specified the HERC4 pulldown in the legend of the Fig.5c.

The authors performed HERC4 pulldown experiments with cell overexpressing three STING variants, 141-341, 155-341 and K150R. Results reported in Extended Data Fig. 6 (a). They observed pull down of all three STING variants and claim that is evidence that the connector helix and K150 are essential for ubiquitination but not ternary complex formation. The flaw with that claim is that there are significant differences in the amount of STING protein pulled down between those three variants.

We thank Reviewer 3 for pointing out this issue. We have now run the same experiment in the presence of bortezomib treatment and provided the evidence of the pulldown of STING constructs in the presence of AK59. We also added DMSO controls of each condition for comparison (Extended data Fig.7e).

The most convincing data demonstrating the AK59 induced interaction between HERC4 and STING are the data reported in Fig. 5(e) from the NanoBiT luciferase reporter assay.

Materials and Methods

All reagents included in the materials and methods section should include both manufacturer and catalog number information.

March 12th, 2024

We thank Reviewer 3 for reminding us this point. We have now added all the catalog numbers, batch numbers of the materials.

Proteomics and analysis: The authors should include the number of cells that were used in the proteomics prep, the quantity of labeled peptides used and how they were detected, the LC gradient used to separate the peptides, and details on how the data analysis was performed (method for scaling the normalization, indications of any missing values in the data, how they dealt with any missing values and the method of imputation). The in-house R (v.3.6) script used to quantify proteins needs to be deposited in Github prior to publication and a citation to that code included in the manuscript.

We thank Reviewer 3 for pointing this out. We have now extended the methods section as well as provided the analysis pipeline. Additionally, we deposited the analysis in Github (https://github.com/Novartis/px_tmt_daa).

Western blotting and co-immunoprecipitation: On line 431, the authors should indicate the concentration and volume of protein loaded.

We thank Reviewer 3 for this comment and we have now added this information in methods.

Figures and Figure legends

Compound concentration used should be added to the legend for Extended Data Fig. 1(b).

We thank Reviewer 3 for this comment and provided this information to the legend.

The legend of Extended Data Fig. 1(d) should indicate what the red dots correspond to.

We thank Reviewer 3 for pointing this out. This information is already provided in supplementary table 1.

The legend of Extended Data Fig. 6(c) should include the time point at which the luminescence was read.

We thank Reviewer 3 for this detail. We have now added this information to the legend.

The label of the Y-axis of Figure 2(c) should indicate the concentration of AK59 used in the experiment.

We thank Reviewer 3 for this point. We have now added this information to the legend.

Figure 2(d) should be edited so that the upper bands of lanes 1 and 2 are not cut off and it is clear that the same bands are not present in lanes 3-8.

We thank Reviewer 3 for pointing out this detail. Unfortunately, these constructs have unspecific band patterns which can also be visible in source data.

To clarify what is represented in the plot, the Y-axis of Fig. 5(a) should be labeled “ Δ GFP signal STING¹⁵⁵⁻³⁴¹ - STING¹⁴¹⁻³⁴¹” and the X-axis should be labeled “time after AK59 treatment”.

We thank Reviewer 3 for this point. We have now changed the Y-axis accordingly. Additionally these data now inverted (e.g. STING¹⁴¹⁻³⁴¹-STING¹⁵⁵⁻³⁴¹) to align with the downward trend of degradation. This was an useful suggestion from Reviewer 1.

March 12th, 2024

Appropriate use of statistics and treatment of uncertainties:

The authors should report EC50's and percent maximal inhibition for the dose response data shown in Extended Data Fig. 1 and Extended Data Fig. 5.

We thank Reviewer 3 for this comment. We added this information to the legend of the corresponding figures.

The authors used the appropriate statistical analyses of the CRISPR screening data (RSA, Q3) and for the identification of significantly enriched or depleted genes in the study groups (Euclidean distance between R,Q coordinates).

The authors should add to the figure legends of Fig. 2(f), 4(j), 5(e) Ext. Data Fig. 1(a), Ext. Data Fig. 5(c), Ext. Data Fig. 6(c) and (d) the p-values that correspond to the asterisk annotations.

We thank Reviewer 3 for pointing out. We have added the p-value asterisk annotations to the legends of the corresponding figures.

The Y-axis of Extended Data Fig. 4 does not include the range of cell count values for each panel.

We thank Reviewer 3 for this point. This graph plotted with the default setting in flowjo by offsetting histogram of each sample. Each curve is already in the same scale. We have now added the count scale for each plot on y-axis.

The X-axis of Extended Data Fig. 1 should indicate which concentration corresponds to the tick mark on the far right.

We thank Reviewer 3 for this detail. We now indicated the concentration in the figure as well as in the legend.

Conclusions:

As detailed in previous sections, there are several conclusions that are not supported by the data included. The conclusions in question are mostly supportive and could be bolstered by further experiments or removed without significantly weakening the central claims.

We thank the reviewer 3 for this critical comment. We have now provided additional data to support the main conclusions of our manuscript as well as added honest criticism of our data to the discussion.

Suggested improvements:

The additional experiments that would significantly contribute to the work are experiments to demonstrate the functional effect of AK59 induced degradation of STING. Those could include demonstrating that AK59 reduces levels of IFN- β , CXCL10 and IL-6 in cGAMP stimulated THP-1 cells. They could also demonstrate that AK59 treatment decreases relative levels of phosphorylated TBK1.

We thank Reviewer 3 for pointing out this aspect. We have now added HTRF assay for cytokine measurements and several Western blots indicating phosphorylation of IRF3 and TBK1 upon cGAMP stimulated THP1 cells in treatment and/or knockout background. These data can be found in Extended Data Fig 1 and extended data Fig. 5.

References:

March 12th, 2024

The authors should add references that support the statements on lines 33 and 35 and for the method cited on line 503. References 16-18 are not cited in the body of manuscript or in the figure legends. Reference 42 is a duplicate of reference 38.

We thank Reviewer 3 for pointing out these mistakes. We have corrected them.

Clarity and context:

The abstract is clear and accessible. The abstract, introduction and discussion sections are appropriate.

Inflammatory material: The manuscript does not contain any language that is inappropriate or potentially libelous.

We thank Reviewer 3 for their critical review of our paper and their insightful suggestions.

Thank you again for all your comments and feedback.

Sincerely,

Merve Mutlu-Koch

Merve Mutlu-Koch, PhD

Discovery Fellow

M +41 79 173 4339

merve.koch@novartis.com

Novartis Institute of Biomedical Research
Discovery Sciences (DSc)

Novartis Campus
Fabrikstrasse 22
WSJ-355/4/005.6
CH-4002 Basel
Switzerland

References

1. Düring, J. *et al.* Structural mechanisms of autoinhibition and substrate recognition by the ubiquitin ligase HACE1. doi:10.1038/s41594-023-01203-4.
 2. Sluimer, J. & Distel, - Ben. Regulating the human HECT E3 ligases. **75**, 3121–3141 (2018).
 3. Lorenz, S. Structural mechanisms of HECT-type ubiquitin ligases. *Biol Chem* **399**, 127–145 (2018).
-

REVIEWERS' COMMENTS

Reviewer #1 (Remarks to the Author):

I would like to thank the authors for a thorough revision and for addressing my questions. I fully support the publication.

Reviewer #2 (Remarks to the Author):

The manuscript has been considerably improved since the initial submission. I agree with most of the comments provided by other reviewers. The responses from the authors, new experiments provided and new text of the manuscript have addressed my previous concerns. The quality of the data and manuscript significantly improved.

Reviewer #3 (Remarks to the Author):

The authors have addressed the majority of suggested revisions.

They have performed additional experiments demonstrating that AK59 treatment inhibits downstream pathway activation by experimentally showing decreases in pTBK1, pIRF3, CXCL10 and IFN β . This data significantly bolsters the authors central hypothesis.

They have added more information about the screening process used to identify and validate AK59 which increases the value of their results to the scientific community.

They determined the discrepancy between my results when searching the STRING database and theirs. They contacted the STRING database help center and determine that it was caused by a version upgrade. I appreciate their efforts.

The only response that wasn't addressed was the request that EC50's and percent maximal inhibition be reported for Extended Fig. 1 and Extended Fig. 5. In their response the authors indicate that those were added to the figure legends but I was unable to locate them. It would be helpful to add those.

April 13th, 2024

Dear reviewers,

Thank you very much for your thorough and constructive feedback. Please find below a point-by-point response of your comments. Our responses are highlighted in blue.

Reviewer #1 (Remarks to the Author):

I would like to thank the authors for a thorough revision and for addressing my questions. I fully support the publication.

We thank reviewer 1 for their support and their prior points that helped us improve our manuscript significantly.

Reviewer #2 (Remarks to the Author):

The manuscript has been considerably improved since the initial submission. I agree with most of the comments provided by other reviewers. The responses from the authors, new experiments provided and new text of the manuscript have addressed my previous concerns. The quality of the data and manuscript significantly improved.

We thank reviewer 2 for their support and their prior points that helped us improve our manuscript significantly

Reviewer #3 (Remarks to the Author):

The authors have addressed the majority of suggested revisions.

They have performed additional experiments demonstrating that AK59 treatment inhibits downstream pathway activation by experimentally showing decreases in pTBK1, pIRF3, CXCL10 and IFN β . This data significantly bolsters the authors central hypothesis.

They have added more information about the screening process used to identify and validate AK59 which increases the value of their results to the scientific community.

They determined the discrepancy between my results when searching the STRING database and theirs. They contacted the STRING database help center and determine that it was caused by a version upgrade. I appreciate their efforts.

April 13th, 2024

The only response that wasn't addressed was the request that EC50's and percent maximal inhibition be reported for Extended Fig. 1 and Extended Fig. 5. In their response the authors indicate that those were added to the figure legends but I was unable to locate them. It would be helpful to add those.

We thank reviewer 3 for pointing out this issue. We apologize that we forgot to add EC50s for these data. We have now added the EC50s for data in Extended Fig. 1 and Extended Fig. 5.

Thank you again for all your comments and feedback.

Sincerely,

Merve Mutlu-Koch

Merve Mutlu-Koch, PhD

Discovery Fellow

M +41 79 173 4339

merve.koch@novartis.com

Novartis Institute of Biomedical Research
Discovery Sciences (DSc)

Novartis Campus
Fabrikstrasse 22
WSJ-355/4/005.6
CH-4002 Basel
Switzerland